# xLSTM-Mixer: Multivariate Time Series Forecasting by Mixing via Scalar Memories

## Abstract

Time series data is prevalent across numerous fields, necessitating the development of robust and accurate forecasting models. Capturing patterns both within and between temporal and multivariate components is crucial for reliable predictions. We introduce xLSTM-Mixer, a model designed to effectively integrate temporal sequences, joint time-variate information, and multiple perspectives for robust forecasting. Our approach begins with a linear forecast shared across variates, which is then refined by xLSTM blocks. They serve as key elements for modeling the complex dynamics of challenging time series data. xLSTM-Mixer ultimately reconciles two distinct views to produce the final forecast. Our extensive evaluations demonstrate its superior long-term forecasting performance compared to recent state-of-the-art methods. A thorough model analysis provides further insights into its key components and confirms its robustness and effectiveness. This work contributes to the resurgence of recurrent models in time series forecasting.

## 1 Introduction

Time series are an essential data modality ubiquitous in many critical fields of application, such as medicine (Hosseini et al., 2021), manufacturing (Essien & Giannetti, 2020), logistics (Seyedan & Mafakheri, 2020), traffic management (Lippi et al., 2013), finance (Lin et al., 2012), audio processing (Latif et al., 2023), weather modeling (Lam et al., 2023). While significant progress in time series forecasting has been made over the decades, the field is still far from being solved. The regular appearance of yet better models and improved combinations of existing approaches exemplifies this. Further increasing the forecast quality obtained from machine learning models promises a manifold of improvements, such as more accurate medical treatments, higher efficiency in manufacturing and transportation, and higher crop yields.

Historically, recurrent neural networks (RNNs) and their powerful successors were natural choices for deep learning-based time series forecasting (Hochreiter & Schmidhuber, 1997; Cho et al., 2014). Today, large Transformers (Vaswani et al., 2017) are applied extensively to time series tasks, including forecasting. Many improvements to the vanilla architecture have since been proposed, including patching (Nie et al., 2023), decompositions (Zeng et al., 2023), tokenization inversions (Liu et al., 2023). However, some of their limitations are yet to be lifted. For instance, they are inefficient when applied to long sequences due to the cost of the attention mechanism being quadratic in the number of variates and time steps, depending on the specific choice of tokenization. Therefore, recurrent and state space models (SSMs) (Patro & Agneeswaran, 2024) are experiencing a resurgence of interest to overcome such limitations. Specifically, Beck et al. (2024) revisited recurrent models by borrowing insights gained from Transformers applied to many domains, specifically to natural language processing. They propose Extended Long Short-Term Memory (xLSTM) models as a viable alternative to current sequence models.

We propose xLSTM-Mixer[1], a new state-of-the-art method for time series forecasting using recurrent deep learning methods. Specifically, we augment the highly expressive xLSTM architecture with carefully crafted time, variate, and multi-view mixing. These operations regularize the training and limit the model parameters by weight-sharing, effectively improving the learning of features necessary for accurate forecasting. xLSTM-Mixer initially computes a channel-independent linear forecast

---

[1]https://anonymous.4open.science/r/xLSTM-Mixer

Figure 1: The xLSTM-Mixer architecture consists of three stages: (1) An initial NLinear forecast assuming channel independence and performing *time mixing*; (2) subsequent *joint mixing*, which mixes variate and time information through crucial applications of sLSTM blocks in two views; and (3) a final *view mixing*, where the two latent forecast views are reconciled into a coherent final forecast.

shared over the variates. It is then up-projected to a higher hidden dimension and subsequently refined by an xLSTM stack. It performs multi-view forecasting by producing a forecast from the original and reversed up-projected embedding. The powerful xLSTM cells thereby jointly mix time and variate information to capture complex patterns from the data. Both forecasts are eventually reconciled by a learned linear projection into the final prediction, again by mixing time. An overview of our method is shown in Fig. 1.

Overall, we make the following contributions:

(i) We investigate time and variate mixing in the context of recurrent models and propose a joint multistage approach that is highly effective for multivariate time series forecasting. We argue that marching over the variates instead of the temporal axis yields better results if suitably combined with temporal mixing.

(ii) We propose xLSTM-Mixer, a state-of-the-art method for time series forecasting using recurrent deep learning methods.

(iii) We extensively compare xLSTM-Mixer with existing methods for multivariate long-term time series forecasting and in-depth model analysis. The experiments demonstrate that xLSTM-Mixer consistently achieves state-of-the-art performance in a wide range of benchmarks.

The following work is structured as follows: In the upcoming Sec. 2, we introduce preliminaries to allow us to motivate and explain xLSTM-Mixer in Sec. 3. We then present comprehensive experiments on its effectiveness and inner workings in Sec. 4. We finally review related work in Sec. 5 and close with a conclusion and outlook in Sec. 6.

## 2 BACKGROUND

After introducing the notation used throughout this work, we review xLSTM blocks and discuss leveraging channel mixing or their independence in time series models.

### 2.1 NOTATION

In multivariate time series forecasting, the model is presented with a time series $\boldsymbol{X} = (\boldsymbol{x}_1, \ldots, \boldsymbol{x}_T) \in \mathbb{R}^{V \times T}$ consisting of $T$ time steps with $V$ variates each. Given this context, the forecaster shall predict the future values $\boldsymbol{Y} = (\boldsymbol{x}_{T+1}, \ldots, \boldsymbol{x}_{T+H}) \in \mathbb{R}^{V \times H}$ up to a horizon $H$. A variate (sometimes called a channel) can be any scalar measurement, such as the temperature at a given point, the occupancy of a road, or the oil temperature in a power plant. The measurements are assumed to be carried out jointly, such that the $T + H$ time steps reflect a regularly sampled signal. A time series dataset consists of $N$ such pairs $\left\{ \left( \boldsymbol{X}^{(i)}, \boldsymbol{Y}^{(i)} \right) \right\}_{i \in \{1, \ldots, N\}}$ divided into train, validation, and test portions.

## 2.2 Extended Long Short-Term Memory (xLSTM)

Beck et al. (2024) propose xLSTM architectures consisting of two building blocks, namely the sLSTM and mLSTM modules. To harness the full expressivity of xLSTMs within each step and across the computation sequence, we employ a stack of sLSTM blocks without any mLSTM blocks. The latter are less suited for joint mixing due to their independent treatment of the sequence elements, making it impossible to learn any relationships between them directly. We will continue by recalling how sLSTM cells function.

The standard LSTM architecture of Hochreiter & Schmidhuber (1997) involves updating the cell state $\mathbf{c}_t$ through a combination of input, forget, and output gates, which regulate the flow of information across tokens. sLSTM blocks enhance this by incorporating exponential gating and memory mixing (Greff et al., 2017) to handle complex temporal and cross-variate dependencies more effectively. The sLSTM updates the cell $\boldsymbol{c}_t$ and hidden state $\boldsymbol{h}_t$ using three gates as follows:

$$\boldsymbol{c}_t = \boldsymbol{f}_t \odot c_{t-1} + \boldsymbol{i}_t \odot \boldsymbol{z}_t \qquad \text{cell state} \qquad (1)$$

$$\boldsymbol{n}_t = \boldsymbol{f}_t \cdot \boldsymbol{n}_{t-1} + \boldsymbol{i}_t \qquad \text{normalizer state} \qquad (2)$$

$$\boldsymbol{h}_t = \boldsymbol{o}_t \odot \boldsymbol{c}_t \odot \boldsymbol{n}_t^{-1} \qquad \text{hidden state} \qquad (3)$$

$$\boldsymbol{z}_t = \tanh\!\big(\boldsymbol{W}_z \boldsymbol{x}_t + \boldsymbol{R}_z h_{t-1} + \boldsymbol{b}_z\big) \qquad \text{cell input} \qquad (4)$$

$$\boldsymbol{i}_t = \exp\!\big(\tilde{\boldsymbol{i}}_t - \boldsymbol{m}_t\big) \qquad \tilde{\boldsymbol{i}}_t = \boldsymbol{W}_i \boldsymbol{x}_t + \boldsymbol{R}_i h_{t-1} + \boldsymbol{b}_i \qquad \text{input gate} \qquad (5)$$

$$\boldsymbol{f}_t = \exp\!\big(\tilde{\boldsymbol{f}}_t + \boldsymbol{m}_{t-1} - \boldsymbol{m}_t\big) \qquad \tilde{\boldsymbol{f}}_t = \boldsymbol{W}_f \boldsymbol{x}_t + \boldsymbol{R}_f h_{t-1} + \boldsymbol{b}_f \qquad \text{forget gate} \qquad (6)$$

$$\boldsymbol{o}_t = \sigma\!\big(\boldsymbol{W}_o \boldsymbol{x}_t + \boldsymbol{R}_o h_{t-1} + \boldsymbol{b}_o\big) \qquad \text{output gate} \qquad (7)$$

$$\boldsymbol{m}_t = \max\!\big(\tilde{\boldsymbol{f}}_t + \boldsymbol{m}_{t-1}, \tilde{\boldsymbol{i}}_t\big) \qquad \text{stabilizer state} \qquad (8)$$

In this setup, the matrices $\boldsymbol{W}_z, \boldsymbol{W}_i, \boldsymbol{W}_f$, and $\boldsymbol{W}_o$ are input weights mapping the input token $\boldsymbol{x}_t$ to the cell input $\boldsymbol{z}_t$, input gate, forget gate, and output gate, respectively. The states $\boldsymbol{n}_t$ and $\boldsymbol{m}_t$ serve as necessary normalization and training stabilization, respectively.

As Beck et al. have shown, it is beneficial to restrict the memory mixing performed by the recurrent weight matrices $\boldsymbol{R}_z, \boldsymbol{R}_i, \boldsymbol{R}_f$, and $\boldsymbol{R}_o$ to individual *heads*, inspired by the multi-head setup of Transformers (Zeng et al., 2023), yet more restricted and efficient. In particular, each token gets broken up into separate pieces, where the input weights $\boldsymbol{W}_{z,i,f,o}$ act across all of them, but the recurrence matrix $\boldsymbol{R}_{z,i,f,o}$ is implemented as block-diagonal and therefore only acts within each piece. This permits specialization of the individual heads to patterns specific to the respective section of the tokens and empirically does not sacrifice expressivity.

## 2.3 Channel Independence and Mixing in Time Series Models

Multiple works have investigated whether it is beneficial to learn representations of the time and variate dimensions jointly or separately. Intuitively, because joint mixing is strictly more expressive, one might think it should always be preferred and is indeed used by many works such as Temporal Convolutional Networks (TCN) (Lea et al., 2016), N-BEATS (Oreshkin et al., 2019), N-HiTS (Challu et al., 2023), and many Transformers (Vaswani et al., 2017), including Temporal Fusion Transformer (TFT) (Lim et al., 2021), Autoformer (Wu et al., 2021), and FEDFormer (Zhou et al., 2022). However, treating slices of the input data independently assumes an invariance to temporal or variate positions and serves as a strong regularization against overfitting, reminiscent of kernels in CNNs. Prominent models implementing some aspects of channel independence in multivariate time series forecasting are PatchTST (Nie et al., 2023) and iTransformer (Liu et al., 2023). TiDE (Das et al., 2023), on the other hand, contains a time-step shared feature projection and temporal decoder but treats variates jointly. As Tolstikhin et al. (2021) have shown with MLP-Mixer, interleaving mixing of all channels in each token and all tokens per channel does not empirically sacrifice any expressivity and instead improves performance. This idea has since been applied to time series too, namely in architectures such as TimeMixer and TSMixer (Chen et al., 2023c), and is one key component of our method xLSTM-Mixer.

## 3 xLSTM-Mixer

We now explain xLSTM-Mixer shown in Fig. 1 in more detail. It carefully integrates several key components: an initial linear forecast with time mixing, joint mixing using powerful sLSTM modules, and an eventual combination of two views by a final fully connected layer. The transposing steps between the components enable capturing complex temporal and intra-variate patterns while facilitating easy trainability and limiting parameter counts. The sLSTM block, in particular, can learn intricate non-linear relationships hidden within the data along both the time and variate dimensions. The architecture is furthermore equipped with normalization layers and skip connections to improve training stability and overall effectiveness.

### 3.1 NORMALIZATION AND INITIAL LINEAR FORECAST

Normalization has become an essential ingredient of modern deep learning architectures (Huang et al., 2023). For time series in particular, reversible instance norm (RevIN) (Kim et al., 2022) is a general recipe for improving forecasting performance, where each time series instance is normalized by its mean and variance and furthermore scaled and offset by learnable scalars $\gamma$ and $\beta$:

$$\boldsymbol{x}_t^{\text{norm}} = \text{RevIN}(\boldsymbol{x}_t) = \gamma \left( \frac{\boldsymbol{x}_t - \mathbb{E}\left[\boldsymbol{x}\right]}{\sqrt{\text{Var}\left[\boldsymbol{x}\right] + \epsilon}} \right) + \beta.$$

We apply it as part of xLSTM-Mixer, and at the end of the entire pipeline, we invert the RevIN operation to obtain the final prediction. In the case of xLSTM-Mixer, the typical skip connections found in mixer arcrhitectures (Tolstikhin et al., 2021; Chen et al., 2023c) are taken up by RevIN, the normalization in the NLinear forecast, and the integral skip connections within each sLSTM block.

It has been shown previously that simple linear models equipped with appropriate normalization schemes are, already by themselves, decent long-term forecasters (Zeng et al., 2023; Li et al., 2023). Our observations confirm this finding. Therefore, we first process each variate separately by an NLinear model by computing:

$$\boldsymbol{x}^{\text{initial}} = \text{NLinear}(\boldsymbol{x}^{\text{norm}}) = \text{FC}\left(\boldsymbol{x}_{1:T}^{\text{norm}} - x_T^{\text{norm}}\right) + x_T^{\text{norm}},$$

where $\text{FC}(\cdot) : \mathbb{R}^T \to \mathbb{R}^H$ denotes a fully-connected linear layer with bias term. Sharing this model across variates limits parameter counts, and the weight-tying serves as a useful regularization. The quality of this initial forecast will be investigated in Sec. 4.1 and 4.2.

### 3.2 SLSTM REFINEMENT

While the NLinear forecast $\boldsymbol{x}^{\text{initial}} \in \mathbb{R}^{V \times H}$ captures the basic patterns between the historic and future time steps, its quality alone is insufficient for today's challenging time series datasets. We, therefore, refine it using powerful sLSTM blocks. As a first step, it is crucial to increase the embedding dimension of the data to provide enough latent dimensions $D$ for the sLSTM cells: $\boldsymbol{x}^{\text{up}} = \text{FC}^{\text{up}}\left(\boldsymbol{x}^{\text{initial}}\right)$. This prior up-projection is similar to what is commonly performed in SSMs (Beck et al., 2024). We weight-share $\text{FC}^{\text{up}} : \mathbb{R}^H \to \mathbb{R}^D$ across variates to perform time-mixing similar to the initial forecast. Note that this step does not maintain the temporal ordering within the embedding token dimensions, as was the case up until this step, and instead embeds it into a higher latent dimension.

The stack of $M$ sLSTM blocks $\mathcal{S}(\cdot)$ transforms $\boldsymbol{x}^{\text{up}} \in \mathbb{R}^{V \times D}$ as defined in Eq. 1 to 8. The recurrent models' strides of the data in variate order, i.e., where each token represents all time steps from a single variate as in the work of Liu et al. (2023). The sLSTM blocks learn intricate non-linear relationships hidden within the data along both the time and variate dimensions. The mixing of the hidden state is still limited to blocks of consecutive dimensions, aiding efficient learning and inference while allowing for effective cross-variate interaction during the recurrent processing. Striding over variates has the benefit of linear time scaling in the number of variates at a constant number of parameters. It, however, comes at the cost of possibly fixing a suboptimal order of variates. While this is empirically not a significant limitation, we leave investigations into how to find a suitable ordering for future work. In addition to a large embedding dim, we observed a high number of heads being crucial for effective forecasting.

Table 1: The long-term forecasting benchmark datasets and their key properties.

| Dataset | Source | Domain | Horizons | Sampling | #Variates | Hurst exp. |
|---------|--------|--------|----------|----------|-----------|------------|
| Weather | Zhou et al. (2021) | Weather | 96–720 | 10 min | 21 | 0.333–1.000 |
| Electricity | Zhou et al. (2021) | Power Usage | 96–720 | 1 hour | 321 | 0.555–1.000 |
| Traffic | Wu et al. (2021) | Traffic Load | 96–720 | 1 hour | 862 | 0.162–1.000 |
| ETT | Zhou et al. (2021) | Power Production | 96–720 | 15&60 min | 7 | 0.906–1.000 |
| Illness (ILI) | Wu et al. (2021) | Influenza cases | 24–60 | 1 week | 7 | 0.499–0.907 |

The sLSTM cells' first hidden state $h_{t-1}$ must be initialized before each sequence of tokens can be processed. Extending the initial description of these blocks, we propose learning a single initial embedding token $\eta \in \mathbb{R}^D$ that gets prepended to each encoded time series $x^{up}$. These initial embeddings draw from recent advances in Large Language Models, where learnable "soft prompt" tokens are used to condition models and improve their ability to generate coherent outputs (Lester et al., 2021; Li & Liang, 2021; Chen et al., 2023a;b). Recent research has extended the application of soft prompts to LLM-based time series forecasting (Cao et al., 2023; Sun et al., 2024), emphasizing their adaptability and effectiveness in improving model performance across modalities. These tokens enable greater flexibility and conditioning, allowing the model to adapt its initial memory representation to specific dataset characteristics and to dynamically interact with the time and variate data. Soft prompts can be readily optimized through back-propagation with very little overhead.

### 3.3 MULTI-VIEW MIXING

To further regularize the training of the sLSTM as with the linear projections, we compute forecasts from the original embedding $x^{up}$ as well as the reversed embedding $\hat{x}^{up}$, where the order of the latent dimensions including the representation of $\eta$ is inverted. Learning forecasts $y', y'' \in \mathbb{R}^{V \times D}$ for both views while sharing weights helps learn better representations. Such multi-task learning settings are known to benefit training (Zhang & Yang, 2022). The final forecast is obtained by a linear projection $\mathrm{FC}^{view} : \mathbb{R}^D \to \mathbb{R}^H$ of the two forecasts, again per-variate. Specifically, we compute:

$$y^{norm} = \mathrm{FC}^{view}(y', y''), \quad \text{where } y' = \mathcal{S}(x^{up}) \text{ and } y'' = \mathcal{S}(\hat{x}^{up}).$$

The final forecast is obtained after de-normalizing the reconciled forecasts as $y = \mathrm{RevIN}^{-1}(y^{norm})$.

## 4 EXPERIMENTS

We conduct a series of experiments to evaluate the forecasting capabilities of xLSTM-Mixer, aiming to provide comprehensive insights into its performance. Our primary focus is on long-term forecasting, following the work of Das et al. (2023); Chen et al. (2023c). An evaluation of xLSTM-Mixer's competitiveness in short-term forecasting on the PEMS dataset is provided in App. A.2. Additionally, we perform an extensive model analysis consisting of an ablation study to identify the contributions of individual components of xLSTM-Mixer, followed by an inspection of the initial embedding tokens, a hyperparameter sensitivity analysis, and an investigation into its robustness.

**Datasets.** We generally follow the established benchmark procedure of Wu et al. (2021) and Zhou et al. (2021) for best backward and future comparability. The datasets we thus used are provided as an overview in Tab. 1. The last column shows the range of Hurst exponents (Hurst, 1951) over the variates measuring long-term patterns. **Training.** We follow standard practice in the forecasting literature by evaluating long-term forecasts using mean squared error (MSE) and mean absolute error (MAE). Based on our experiments, we used MAE as the training loss function since it yielded the best results. The datasets were standardized for consistency across features. In addition, we conducted all experiments three times and reported the averaged values. Further details on hyperparameter selection, metrics, and implementation can be found in App. A.1. **Baseline Models.** We compare xLSTM-Mixer to the recurrent models xLSTMTime (Alharthi & Mahmood, 2024) and LSTM (Hochreiter & Schmidhuber, 1997); multi-perceptron (MLP) based models TimeMixer (Wang et al., 2024a), TSMixer (Chen et al., 2023c), DLinear (Zeng et al., 2023), and TiDE (Das et al., 2023); the transformers PatchTST (Nie et al., 2023), iTransformer (Liu et al., 2023), FEDFormer (Zhou et al., 2022), and Autoformer (Wu et al., 2021); and the convolutional architectures MICN (Wang et al., 2022) and TimesNet (Wu et al., 2022).

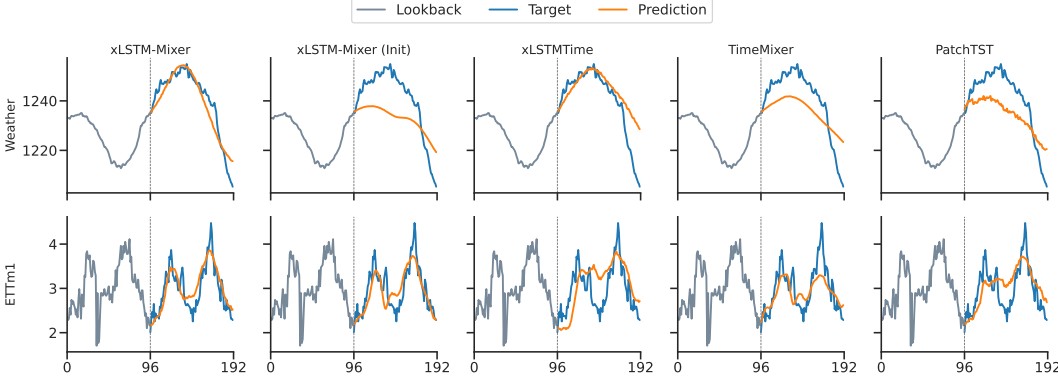

Figure 2: **Example Forecasts Across Models and Datasets.** This figure shows example forecasts on the Weather and ETTm1 datasets for multiple models. The lookback window and forecasting horizon are fixed at 96. The blue lines represent the ground truth target, and the orange lines show the predictions. The first panel illustrates the forecast from xLSTM-Mixer, while the second panel shows the forecast extracted before the up-projection step, highlighting the effectiveness of our added components. Comparisons with xLSTMTime, TimeMixer, and PatchTST provide insights into the performance of xLSTM-Mixer relative to these baseline models.

Table 2: **xLSTM-Mixer is effective in long-term forecasting.** The results are averaged from 4 different prediction lengths {96, 192, 336, 720}, and {24, 36, 48, 60} for Ili. A lower MSE or MAE indicates a better prediction. The **best** result for each dataset is highlighted in bold red, while the second-best result is blue and underlined. Wins for each model out of all 32 settings are shown at the bottom.

| Models | Recurrent | | | | | | MLP | | | | | | | | Transformer | | | | | | | | Convolutional | | | |
|---|---|---|---|---|---|---|---|---|---|---|---|---|---|---|---|---|---|---|---|---|---|---|---|---|---|---|---|
| | xLSTM-Mixer (Ours) | | xLSTMTime 2024 | | LSTM 1997[a] | | TimeMixer 2024a | | TSMixer 2023c | | DLinear 2023 | | TiDE 2023 | | PatchTST 2023 | | iTransformer 2023 | | FEDFormer 2022 | | Autoformer 2021 | | MICN 2022 | | TimesNet 2022 | |
| Dataset | MSE | MAE | MSE | MAE | MSE | MAE | MSE | MAE | MSE | MAE | MSE | MAE | MSE | MAE | MSE | MAE | MSE | MAE | MSE | MAE | MSE | MAE | MSE | MAE | MSE | MAE |
| Weather | **0.219** | **0.250** | 0.222 | 0.255 | 0.444 | 0.454 | 0.222 | 0.262 | 0.225 | 0.264 | 0.246 | 0.300 | 0.236 | 0.282 | 0.241 | 0.264 | 0.258 | 0.278 | 0.309 | 0.360 | 0.338 | 0.382 | 0.242 | 0.299 | 0.259 | 0.287 |
| Electricity | **0.153** | **0.245** | 0.157 | 0.250 | 0.559 | 0.549 | 0.156 | 0.246 | 0.160 | 0.256 | 0.166 | 0.264 | 0.159 | 0.257 | 0.159 | 0.253 | 0.178 | 0.270 | 0.214 | 0.321 | 0.227 | 0.338 | 0.186 | 0.295 | 0.192 | 0.295 |
| Traffic | 0.392 | 0.253 | 0.391 | 0.261 | 1.011 | 0.541 | 0.387 | 0.262 | 0.408 | 0.284 | 0.434 | 0.295 | **0.356** | 0.261 | 0.391 | 0.264 | 0.428 | 0.282 | 0.609 | 0.376 | 0.628 | 0.379 | 0.541 | 0.315 | 0.620 | 0.336 |
| Illness | **1.426** | 0.719 | 1.488 | **0.714** | 6.538 | 1.829 | 1.747 | 0.829 | 2.476 | 1.060 | 2.616 | 1.090 | 1.899 | 0.934 | 1.480 | 0.807 | 1.845 | 0.906 | 2.847 | 1.144 | 3.006 | 1.161 | 2.664 | 1.086 | 2.139 | 0.931 |
| ETTh1 | **0.397** | **0.420** | 0.408 | 0.428 | 1.198 | 0.821 | 0.411 | 0.423 | 0.412 | 0.428 | 0.423 | 0.437 | 0.419 | 0.430 | 0.413 | 0.434 | 0.454 | 0.448 | 0.440 | 0.460 | 0.496 | 0.487 | 0.558 | 0.535 | 0.458 | 0.450 |
| ETTh2 | 0.340 | 0.382 | 0.346 | 0.386 | 3.095 | 1.352 | **0.316** | 0.384 | 0.355 | 0.401 | 0.431 | 0.447 | 0.345 | 0.394 | 0.324 | **0.381** | 0.383 | 0.407 | 0.433 | 0.447 | 0.453 | 0.462 | 0.588 | 0.525 | 0.414 | 0.427 |
| ETTm1 | **0.339** | **0.366** | 0.347 | 0.372 | 1.142 | 0.782 | 0.348 | 0.375 | 0.347 | 0.375 | 0.357 | 0.379 | 0.355 | 0.378 | 0.353 | 0.382 | 0.407 | 0.410 | 0.448 | 0.452 | 0.588 | 0.517 | 0.392 | 0.413 | 0.400 | 0.406 |
| ETTm2 | **0.248** | **0.307** | 0.254 | 0.310 | 2.395 | 1.177 | 0.256 | 0.315 | 0.267 | 0.322 | 0.267 | 0.332 | 0.249 | 0.312 | 0.256 | 0.317 | 0.288 | 0.332 | 0.304 | 0.349 | 0.324 | 0.368 | 0.328 | 0.382 | 0.291 | 0.333 |
| Wins | **20** | **25** | 2 | 4 | 0 | 0 | 3 | 5 | 0 | 0 | 0 | 0 | 5 | 1 | 2 | 1 | 0 | 0 | 0 | 0 | 0 | 0 | 0 | 0 | 0 | 0 |

[a] Taken from Wu et al. (2022).

## 4.1 Long-Term Time Series Forecasting

We present the performance of xLSTM-Mixer compared to prior models in Tab. 2. As shown, xLSTM-Mixer consistently delivers highly accurate forecasts across a wide range of datasets. It achieves the best results in 20 out of 32 cases for MSE and 25 out of 32 cases for MAE, demonstrating its superior performance in long-term forecasting. In particular, xLSTM-Mixer exhibits exceptional forecasting accuracy, as evidenced particularly by its strong MAE performance across all datasets. Notably, on Weather, xLSTM-Mixer reduces the MAE by 2% compared to xLSTMTime and 4.8% compared to TimeMixer. Similarly, for ETTm1, xLSTM-Mixer outperforms TimeMixer by 2.46% in MAE and shows a strong competitive edge over xLSTMTime. Although xLSTM-Mixer performs slightly less well on the Traffic and ETTh2 datasets, where it encounters challenges with handling outliers, it remains highly competitive and outperforms the majority of baseline models. This suggests that despite these few cases, xLSTM-Mixer can consistently deliver state-of-the-art performance in long-term forecasting. A qualitative inspection of several baseline models, including the initial forecast extracted before the sLSTM refinement, is shown in Fig. 2. In this comparison, the lookback window and forecasting horizon are both fixed at 96.

Table 3: **Ablation Study on the Weather and ETTm1 Datasets.** MSE and MAE are reported for each numbered configuration across four prediction lengths. The **best** results are highlighted in bold red, while the second-best results are blue and underlined.

| | | #1 (full) | #2 | #3 | #4 | #5 | #6 | #7 | #8 | #9 | #10 | #11 | #12 |
|---|---|---|---|---|---|---|---|---|---|---|---|---|---|
| | Time Mixing | ✓ | ✓ | ✓ | DLinear | ✓ | ✓ | ✓ | ✓ | ✗ | ✗ | ✗ | ✗ |
| | xLSTM type | sLSTM | mLSTM | sLSTM | sLSTM | sLSTM | sLSTM | sLSTM | None | sLSTM | sLSTM | sLSTM | sLSTM |
| | Recurr. order | Variates | Variates | Time | Variates | Variates | Variates | Variates | None | Variates | Variates | Variates | Variates |
| | Init. Token | ✓ | ✓ | ✓ | ✓ | ✗ | ✓ | ✗ | ✗ | ✓ | ✗ | ✓ | ✗ |
| | View Mixing | ✓ | ✓ | ✓ | ✓ | ✓ | ✗ | ✗ | ✗ | ✓ | ✓ | ✗ | ✗ |
| | Metric | MSE MAE | MSE MAE | MSE MAE | MSE MAE | MSE MAE | MSE MAE | MSE MAE | MSE MAE | MSE MAE | MSE MAE | MSE MAE | MSE MAE |
| Weather | 96 | **0.143 0.184** | 0.148 0.192 | 0.148 0.194 | 0.145 0.187 | 0.145 0.186 | 0.144 0.185 | 0.144 0.186 | 0.173 0.223 | 0.149 0.193 | 0.151 0.195 | 0.149 0.192 | 0.152 0.195 |
| | 192 | **0.186 0.226** | 0.193 0.235 | 0.196 0.239 | 0.188 0.229 | 0.188 0.228 | 0.186 0.226 | 0.188 0.228 | 0.219 0.257 | 0.192 0.233 | 0.192 0.234 | 0.191 0.234 | 0.193 0.236 |
| | 336 | **0.237 0.266** | 0.241 0.272 | 0.252 0.281 | 0.237 0.267 | 0.239 0.267 | 0.241 0.270 | 0.242 0.270 | 0.261 0.288 | 0.240 0.271 | 0.242 0.273 | 0.242 0.273 | 0.244 0.274 |
| | 720 | 0.310 0.324 | 0.313 0.325 | 0.315 0.328 | 0.312 0.325 | 0.310 0.324 | 0.309 0.323 | 0.309 0.323 | 0.320 0.334 | 0.320 0.329 | 0.319 0.329 | 0.322 0.330 | 0.319 0.328 |
| ETTm1 | 96 | 0.275 0.328 | 0.285 0.339 | 0.298 0.348 | 0.274 0.329 | 0.277 0.329 | 0.278 0.331 | 0.279 0.333 | 0.295 0.338 | 0.282 0.339 | 0.285 0.341 | 0.281 0.337 | 0.284 0.339 |
| | 192 | 0.319 0.354 | 0.329 0.365 | 0.337 0.369 | 0.319 0.356 | 0.321 0.354 | 0.321 0.356 | 0.322 0.358 | 0.329 0.357 | 0.329 0.364 | 0.330 0.365 | 0.337 0.367 | 0.335 0.366 |
| | 336 | 0.353 0.374 | 0.363 0.384 | 0.368 0.388 | 0.351 0.376 | 0.354 0.375 | 0.355 0.377 | 0.357 0.379 | 0.359 0.376 | 0.367 0.385 | 0.367 0.385 | 0.366 0.384 | 0.366 0.385 |
| | 720 | **0.409 0.407** | 0.417 0.414 | 0.420 0.416 | 0.409 0.408 | 0.411 0.408 | 0.413 0.411 | 0.414 0.411 | 0.412 0.407 | 0.422 0.412 | 0.422 0.413 | 0.417 0.410 | 0.418 0.411 |

## 4.2 MODEL ANALYSIS

**Ablation Study.** To assess the contributions of each component in xLSTM-Mixer to its strong forecast performance, we conducted an ablation study with results listed in Tab. 3. Each configuration represents a different combination of the four key components: mixing time with NLinear and DLinear (D), using sLSTM blocks, learning an initial embedding token, and multi-view mixing. We evaluated the performance using the MSE and MAE across prediction lengths $\{96, 192, 336, 720\}$. The full version of xLSTM-Mixer (#1), which integrates all components, achieves the best performance overall. However, we also observe that some configurations of xLSTM-Mixer, which exclude specific components, remain competitive. For instance, #5, which excludes the initial embedding token, still performs very well. Similarly, depending on the dataset and target metric, initial forecasting with DLinear instead of NLinear is a viable option, too (#4). This suggests that while it contributes positively to the overall performance, the model can sometimes still achieve competitive results without it. In general, removing specific components leads to a performance drop. For example, removing the time mixing (#9) increases the MAE by 3.4% on ETTm1 at length 96 or 3.1% at length 192, highlighting its critical role in capturing cross-time dependencies. When we now omit everything except for time mixing on Weather at 192, we suffer a 13.7% performance decrease. In summary, the ablation study confirms that all components of xLSTM-Mixer contribute to its effectiveness, with the full configuration yielding the best results. Furthermore, we identified the sLSTM blocks and time-mixing components as critical for ensuring high accuracy across datasets and prediction lengths.

**Sensitivity to xLSTM Hidden Dimension.** In Fig. 4, we visualize the performance of xLSTM-Mixer on the Electricity dataset with increasing sLSTM embedding (hidden) dimension realized by $\text{FC}^{\text{up}}$. The results indicate that larger hidden dimensions consistently enhance the model's performance, particularly for longer prediction lengths. This suggests that a larger embedding dimension enables xLSTM-Mixer to better capture the complexity of the time series data over extended horizons, leading to improved forecasting accuracy.

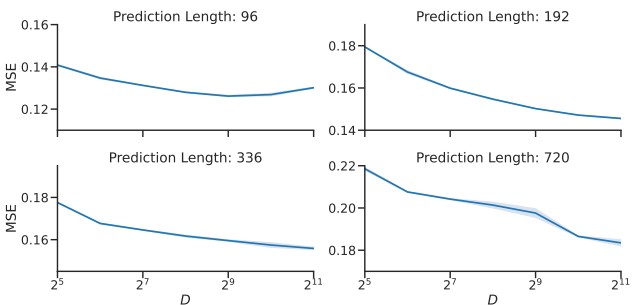

Figure 4: Sensitivity analysis of the xLSTM hidden dimension $D$ on the Electricity dataset. Increasing the up-projection dimension becomes beneficial with increasing prediction length.

**Initial Token Embedding.** We qualitatively inspect decodings of the initial embedding tokens $\boldsymbol{\eta}$ on multiple datasets to further understand and interpret the initializations learned by xLSTM-Mixer. $\boldsymbol{\eta}$ are decoded to a forecast $\boldsymbol{y}$ by transforming them through the sLSTM stack $\mathcal{S}$ and applying multi-view mixing. The resulting output of $\text{FC}^{\text{view}}$ can then be interpreted as the conditioning time

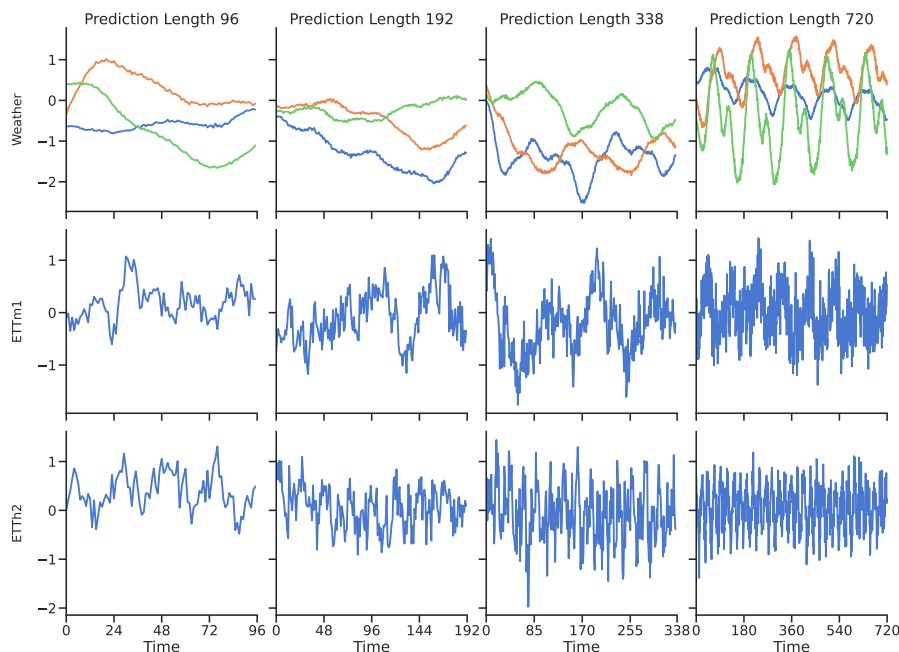

Figure 3: **Initial Tokens Capture Dataset Characteristics.** The plot illustrates the learned tokens across multiple datasets and prediction lengths. The lookback length is set to 96 for all evaluations. For clarity and the high noise levels of the data, only a single seed is shown for ETTm1 and ETTh2.

series used to initialize the sLSTM blocks. Fig. 3 shows the dataset-specific patterns the initial embedding tokens have learned on Weather, ETTm1, and ETTh2 for various prediction horizons. With increasing prediction horizons, we observe longer spans of time, eventually revealing underlying seasonal patterns and respective dataset dynamics.

**Learning Cross-Variate Patterns.** As the sLSTM refinement blocks recurrently process the variates, it is important to assess the extent to which inter-variate relationships are effectively captured. To this end, we adopt a perturbation-based approach to compute attributions, approximating Shapley Values through sampling. Hereby, we use a zero baseline and follow the horizon aggregation method proposed by Kraus et al. (2024), where the forecasts over the entire horizon are aggregated into a single scalar, which serves as the target for the attribution computation. We visualize these Shapley-based feature attribution scores, illustrating the degree to which each output variate of the xLSTM-Mixer depends on each input variate. Fig. 5 demonstrates the ability of xLSTM-Mixer to model cross-variate relationships effectively. Due to the design of the sLSTM refinement module, which strides over the variates, each variate can only be influenced by the ones preceding it. This restriction is reflected in the attribution scores, which appear exclusively in the lower-left triangle. In addition to these results on the Weather dataset, App. A.3 provides results for ETTh2 and ETTm1.

**Model Efficiency.** To survey the computational resources required for using xLSTM-Mixer, we measured the averaged wall-clock time and peak graphics card memory required to perform a training step. Fig. 7 shows how this changes over multiple lookback lengths $T$ and two datasets at a forecast horizon of $H = 336$. We also perform this experiment for NLinear, PatchTST, and TimeMixer to put the measurements into context. xLSTM-Mixer scales very favorably in $T$, exhibiting a negligible increase in time and memory requirements compared to

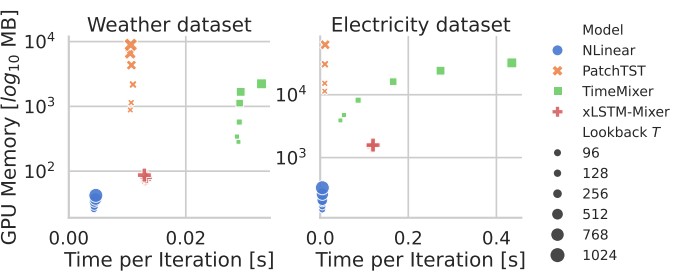

Figure 7: Model efficiency.

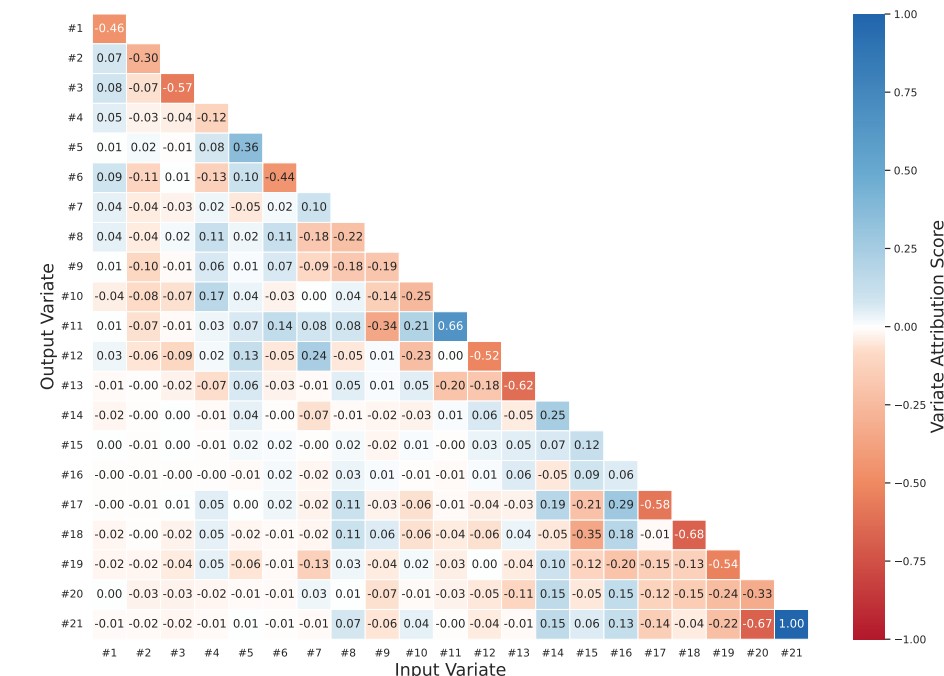

Figure 5: **xLSTM-Mixer effectively learns cross-variate patterns,** as this feature attribution of each output to input variate on the Weather dataset demonstrates.

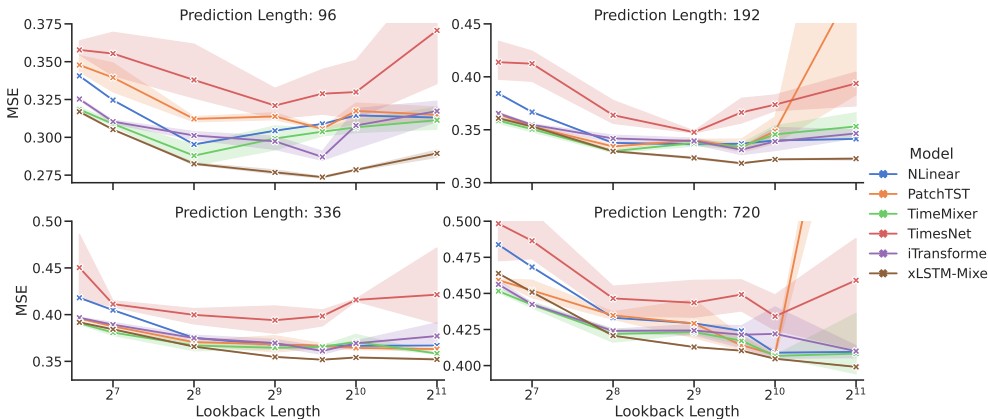

Figure 6: **Increasing the lookback window increases forecasting performance,** with xLSTM-Mixer virtually always providing the best results. Shows the MSE ($\downarrow$) on the ETTm1 dataset.

the other models. While computations take slightly longer for small lookback sizes, the increase is much smaller than for Transformer-based models. One advantage of TimeMixer was its efficiency over Transformers, upon which xLSTM-Mixer now significantly improves by requiring one or two orders of magnitude less memory.

**Robustness to Lookback Length.** Fig. 6 illustrates the performance of xLSTM-Mixer across varying lookback lengths $T$ and prediction horizons $H$. Note that we had to rerun some experiments for TimeMixer at $T = 720$ with varying seeds since many training runs diverged. We observe that xLSTM-Mixer can effectively utilize longer lookback windows than the baselines, especially when compared to transformer-based models. This advantage stems from xLSTM-Mixer's avoidance of self-attention, allowing it to handle extended lookback lengths efficiently. On short prediction lengths with $T \in \{96, 192\}$, information of more than 768 time steps in the past becomes redundant to inform the comparatively short forecast, causing models to deteriorate slightly. On longer horizons, increasingly

farther lookbacks become useful for forecasting. Additionally, xLSTM-Mixer demonstrates stable and consistent performance with low variance across scales.

## 5 RELATED WORK

**Time Series Forecasting.** A long line of machine learning research led from early statistical methods like ARIMA to contemporary models based on deep learning, where four architectural families take center stage: based on recurrence, state spaces, convolutions, Multilayer Perceptrons (MLPs), and Transformers. While all of them are used by practitioners today, the research focus is gradually shifting over time. Initially, the naturally sequential recurrent models such as Long Short-Term Memory (LSTM) (Hochreiter & Schmidhuber, 1997) and Gated Recurrent Units (GRUs) (Cho et al., 2014) were used for time series analysis. Their main benefits are the high inference efficiency and arbitrary input and output lengths due to their autoregressive nature. While their effectiveness has historically been constrained by a limited ability to capture long-range dependencies, active research remains to alleviate these limitations (Salinas et al., 2020), including the xLSTM architecture presented in Sec. 2 (Beck et al., 2024; Alharthi & Mahmood, 2024) and SutraNets (Bergsma et al., 2023). Closely related are state space models (SSMs) such as Mamba (Gu & Dao, 2024; Wang et al., 2024c), which permit parallel inference for improved efficiency. Similarly efficient, yet more restricted in their output length, are the location-invariant CNNs (Li et al., 2022; Lara-Benítez et al., 2021), such as TCN (Lea et al., 2016), TimesNet (Wu et al., 2022), and MICN (Wang et al., 2022). Recently, some MLP-based architectures have also shown good success, including the simplistic DLinear and NLinear models (Zeng et al., 2023), the encoder-decoder architecture of TiDE (Das et al., 2023), the mixing architectures TimeMixer (Wang et al., 2024a) and TSMixer (Chen et al., 2023c), as well as the hierarchical N-BEATS Oreshkin et al. (2019) and N-HiTS (Challu et al., 2023) models. Finally, a lot of models have been proposed based on Transformers (Vaswani et al., 2017), such as Autoformer (Wu et al., 2021), TFT (Lim et al., 2021), FEDFormer (Zhou et al., 2022), PatchTST (Nie et al., 2023), and iTransformer (Liu et al., 2023).

**xLSTM Models for Time Series.** Some initial experiments of applying xLSTMs (Beck et al., 2024) to time series were already performed by Alharthi & Mahmood (2024) with their proposed xLSTMTime model. While it showed promising forecasting performance, these initial soundings did not surpass stronger recent models such as TimeMixer (Wang et al., 2024a) on multivariate benchmarks, and the reported performance is challenging to reproduce. We ensure that our method xLSTM-Mixer is well suited as a foundation for further research by providing extensive model analysis, including an ablation study with ten variants and more, and ensuring that results are readily reproducible. Our methodology draws from xLSTMTime yet improves on it by several key components. Most importantly, our novel multi-view mixing consistently enhances forecasting performance. Furthermore, we find the trend-seasonality decomposition redundant and a simple NLinear normalization scheme (Zeng et al., 2023) to suffice. Concurrently, Kong et al. (2024) also investigate using xLSTMs for time series forecasting, arriving at similar conclusions.

## 6 CONCLUSION

In this work, we introduced xLSTM-Mixer, a method that combines a linear forecast with further refinement using xLSTM blocks. Our architecture effectively integrates time, joint, and view mixing to capture complex dependencies. In long-term forecasting, xLSTM-Mixer consistently achieved state-of-the-art performance, outperforming previous methods in 45 out of 64 cases. Furthermore, our detailed model analysis provided valuable insights into the contribution of each component and demonstrated its robustness to varying hyperparameter settings.

While xLSTM-Mixer has shown extraordinary performance in long-term forecasting, it should be noted that due to the transpose of the input, i.e., processing the variates as sequence elements, the number of variates may limit the overall performance. To overcome this, we plan to explore how different variate orderings influence performance and whether incorporating more than two views could lead to further improvements. This study focused on long-term forecasting, yet extending xLSTM-Mixer to tasks such as short-term forecasting, time series classification, or imputation offers promising directions for future research.

ETHICS STATEMENT

Our research advances machine learning by enhancing the capabilities of long-term forecasting in time series models, significantly improving both accuracy and efficiency. By developing xLSTM-Mixer, we introduce a robust framework that can be applied across various industries, including finance, healthcare, energy, and logistics. The improved forecasting accuracy enables better decision-making in critical areas, such as optimizing resource allocation, predicting market trends, and managing risk.

However, we also recognize the potential risks associated with the misuse of these advanced models. Time series forecasting models could be leveraged for malicious purposes, especially when applied at scale. For example, in the financial sector, adversarial agents might manipulate forecasts to create market instability. In political or social contexts, these models could be exploited to predict and influence public opinion or destabilize economies. Additionally, the application of these models in sensitive domains like healthcare and security may lead to unintended consequences if not carefully regulated and ethically deployed.

Therefore, it is essential that the use of xLSTM-Mixer, like all machine learning technologies, is guided by responsible practices and ethical considerations. We encourage stakeholders to adopt rigorous evaluation processes to ensure fairness, transparency, and accountability in its deployment, and to remain vigilant to the broader societal implications of time series forecasting technologies.

REPRODUCIBILITY STATEMENT

All implementation details, including dataset descriptions, metric calculations, and experiment configurations, are provided in Sec. 4 and App. A.1. We make sure to exclusively use openly available software and datasets and provide the source code for full reproducibility at
`https://anonymous.4open.science/r/xLSTM-Mixer`.

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

# A  APPENDIX

## A.1  IMPLEMENTATION DETAILS

**Experimental Details.**  Our codebase is implemented in Python 3.11, leveraging PyTorch 2.4 (Paszke et al., 2019) in combination with Lightning 2.4[2] for model training and optimization. We used the custom CUDA implementation[3] for sLSTM which has a reliance on the NVIDIA Compute Capability >= 8.0. Thus, our experiments were conducted on a single NVIDIA A100 80GB GPU. The majority of our baseline implementations, along with data loading and preprocessing steps, are adapted from the Time-Series-Library[4] of Wang et al. (2024b). Additionally, for xLSTMTime we used code based on the official repository[5]. We employ Captum[6] (Kokhlikyan et al., 2020) to compute the SHAP values used in model analysis.

**Training and Hyperparameters.**  We optimized xLSTM-Mixer for up to 60 epochs with a cosine-annealing scheduler with the Adam optimizer (Kingma & Ba, 2017), using $\beta_1 = 0.9$ and $\beta_2 = 0.999$ and no weight decay. Hyperparameter tuning was conducted using Optuna (Akiba et al., 2019) with the choices provided in Tab. 4. We optimized for the L1 forecast error, also known as the Mean Absolute Error (MAE). To further stabilize the training process, gradient clipping with a maximum norm of 1.0 was applied. All experiments were run with three different random seeds {2021, 2022, 2023}.

Table 4: Hyperparameters and their choices.

| Hyperparameter | Choices |
|---|---|
| Batch size | $\{16, 32, 64, 128, 256, 512\}$ |
| Initial learning rate | $\{1 \cdot 10^{-2}, 3 \cdot 10^{-3}, 1 \cdot 10^{-3}, 5 \cdot 10^{-4}, 2 \cdot 10^{-4}, 1 \cdot 10^{-4}\}$ |
| Scheduler warmup steps | $\{5, 10, 15\}$ |
| Lookback length | $\{96, 256, 512, 768, 1024, 2048\}$ |
| Embedding dimension $D$ | $\{32, 64, 128, 256, 512, 768, 1024\}$ |
| sLSTM conv. kernel width | $\{\text{disabled}, 2, 4\}$ |
| sLSTM dropout rate | $\{0.1, 0.25\}$ |
| # sLSTM blocks | $\{1, 2, 3, 4\}$ |
| # sLSTM heads | $\{4, 8, 16, 32\}$ |

**Metrics.**  We follow common practice in the literature (Wu et al., 2021; Wang et al., 2024a) for maximum comparability and, therefore, evaluate long-term forecasting of all models on the mean absolute error (MAE), mean squared error (MSE), and for short-term forecasting, using the MAE, root mean squared error (RMSE), and mean absolute percentage error (MAPE). The metrics are averaged over all variates and computed as:

$$\text{MAE}(\boldsymbol{y}, \hat{\boldsymbol{y}}) = \sum_{i=1}^{H} |y_i - \hat{y}_i| \qquad \text{MSE}(\boldsymbol{y}, \hat{\boldsymbol{y}}) = \sum_{i=1}^{H} (y_i - \hat{y}_i)^2$$

$$\text{RMSE}(\boldsymbol{y}, \hat{\boldsymbol{y}}) = \sqrt{\text{MSE}(\boldsymbol{y}, \hat{\boldsymbol{y}})} \qquad \text{MAPE}(\boldsymbol{y}, \hat{\boldsymbol{y}}) = \frac{100}{H} \sum_{i=1}^{H} \frac{|y_i - \hat{y}_i|}{|y_i|},$$

where $\boldsymbol{y}$ are the targets, $\hat{\boldsymbol{y}}$ the predictions, and $\epsilon$ a small constant added for numerical stability.

---

[2]https://lightning.ai/pytorch-lightning
[3]https://github.com/NX-AI/xlstm
[4]https://github.com/thuml/Time-Series-Library
[5]https://github.com/muslehal/xLSTMTime
[6]https://captum.ai

## A.2 OUTLOOK: SHORT-TERM TIME SERIES FORECASTING

Having shown superior long-term forecasting accuracies in Sec. 4.1, we also provide an initial exploration of the effectiveness of xLSTM-Mixer to short-term forecasts. To this end, we compare it to applicable baselines on PEMS datasets with input lengths uniformly set to 96 and prediction lengths to 12. The results in Tab. 5 show that the performance of xLSTM-Mixer is competitive with existing methods. We provide the MAE, MAPE, and RMSE as is common practice.

Table 5: **Short-term forecasting evaluation of xLSTM-Mixer and baselines on the multivariate PEMS datasets.** A lower MAE, MAPE, or RMSE indicates a better prediction. The **best** result for each dataset is highlighted in bold red, while the second-best result is blue and underlined. Wins for each model are shown at the bottom.

| Models | | Recurrent | | | MLP | | Transformer | | | Convolutional | |
| --- | --- | --- | --- | --- | --- | --- | --- | --- | --- | --- | --- |
| | | **xLSTM-Mixer** (**Ours**) | xLSTMTime 2024 | LSTM 1997[a] | TimeMixer 2024a | DLinear 2023 | PatchTST 2023 | FEDFormer 2022 | Autoformer 2021 | MICN 2022 | TimesNet 2022 |
| PEMS03 | MAE | 15.71 | 16.59 | 18.65 | **14.63** | 19.70 | 18.95 | 19.00 | 18.08 | 15.71 | 16.41 |
| | MAPE | 14.92 | 15.31 | 17.39 | **14.54** | 18.35 | 17.29 | 18.57 | 18.75 | 15.67 | 15.17 |
| | RMSE | 24.82 | 26.47 | 31.73 | **23.28** | 32.35 | 30.15 | 30.05 | 27.82 | 24.55 | 26.72 |
| PEMS08 | MAE | 16.56 | 17.44 | 20.34 | **15.22** | 20.26 | 20.35 | 20.56 | 20.47 | 17.76 | 19.01 |
| | MAPE | 10.24 | 10.58 | 13.05 | **9.67** | 12.09 | 13.15 | 12.41 | 12.27 | 10.76 | 11.83 |
| | RMSE | 26.65 | 28.13 | 31.90 | **24.26** | 32.38 | 31.04 | 32.97 | 31.52 | 27.26 | 30.65 |

[a] Configuration following Wu et al. (2021).

## A.3 ADDITIONAL RESULTS ON CROSS-VARIATE PATTERN LEARNING

Fig. 8 extends the experimental findings in Sec. 4.2.

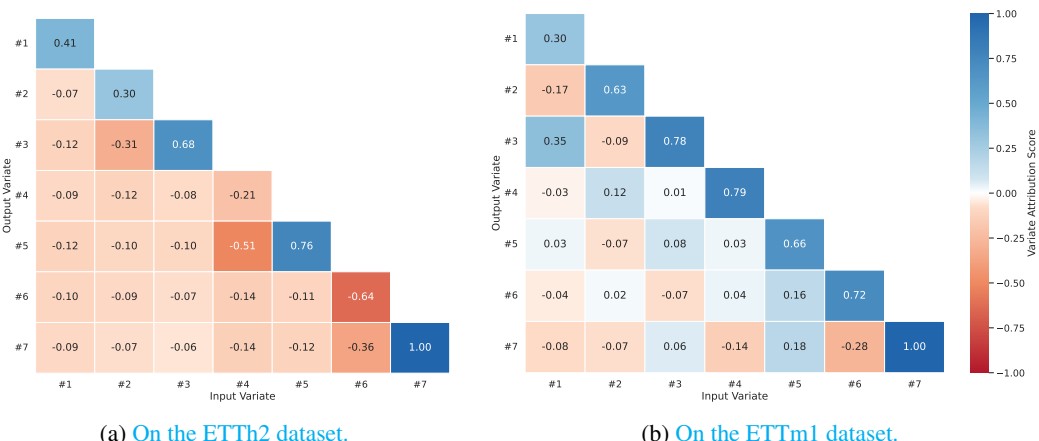

(a) On the ETTh2 dataset.

(b) On the ETTm1 dataset.

Figure 8: **xLSTM-Mixer effectively learns cross-variate patterns.**

## A.4 FULL RESULTS FOR LONG-TERM FORECASTING

Tab. 6 shows the full results for long-term forecasting.

Table 6: **Full long-term forecasting results for Tab. 2.** *Avg* is averaged from all four prediction lengths {96, 192, 336, 720}, and {24, 36, 48, 60} for Ill. A lower MSE or MAE indicates a better prediction. The **best** result for each dataset is highlighted in bold red, while the second-best result is blue and underlined. Wins for each model are shown at the bottom.

| | | Recurrent | | | | | MLP | | | | | | | Transformer | | | | | | | | Convolutional | | |
|---|---|---|---|---|---|---|---|---|---|---|---|---|---|---|---|---|---|---|---|---|---|---|---|---|
| Models | | xLSTM-Mixer (Ours) | | xLSTMTime 2024 | | LSTM 1997[a] | | TimeMixer 2024a | | TSMixer 2023c | | DLinear 2023 | | TiDE 2023 | | PatchTST 2023 | | iTransformer 2023 | | FEDFormer 2022 | | Autoformer 2021 | | MICN 2022 | | TimesNet 2022 | |
| Metric | | MSE | MAE | MSE | MAE | MSE | MAE | MSE | MAE | MSE | MAE | MSE | MAE | MSE | MAE | MSE | MAE | MSE | MAE | MSE | MAE | MSE | MAE | MSE | MAE | MSE | MAE |
| Weather | 96 | 0.143 | 0.184 | 0.144 | 0.187 | 0.369 | 0.406 | 0.147 | 0.197 | 0.145 | 0.198 | 0.176 | 0.237 | 0.166 | 0.222 | 0.149 | 0.198 | 0.174 | 0.214 | 0.217 | 0.296 | 0.266 | 0.336 | 0.161 | 0.229 | 0.172 | 0.220 |
| | 192 | 0.186 | 0.226 | 0.192 | 0.236 | 0.416 | 0.435 | 0.189 | 0.239 | 0.191 | 0.242 | 0.220 | 0.282 | 0.209 | 0.263 | 0.194 | 0.241 | 0.221 | 0.254 | 0.276 | 0.336 | 0.307 | 0.367 | 0.220 | 0.281 | 0.219 | 0.261 |
| | 336 | 0.236 | 0.266 | 0.237 | 0.272 | 0.455 | 0.454 | 0.241 | 0.280 | 0.242 | 0.280 | 0.265 | 0.319 | 0.254 | 0.301 | 0.306 | 0.282 | 0.278 | 0.296 | 0.339 | 0.380 | 0.359 | 0.395 | 0.278 | 0.331 | 0.280 | 0.306 |
| | 720 | 0.310 | 0.323 | 0.313 | 0.326 | 0.535 | 0.520 | 0.310 | 0.330 | 0.320 | 0.336 | 0.323 | 0.362 | 0.313 | 0.340 | 0.314 | 0.334 | 0.358 | 0.347 | 0.403 | 0.428 | 0.419 | 0.428 | 0.311 | 0.356 | 0.365 | 0.359 |
| | Avg | 0.219 | 0.250 | 0.222 | 0.255 | 0.444 | 0.454 | 0.222 | 0.262 | 0.225 | 0.264 | 0.246 | 0.300 | 0.236 | 0.282 | 0.241 | 0.264 | 0.258 | 0.278 | 0.309 | 0.360 | 0.338 | 0.382 | 0.242 | 0.299 | 0.259 | 0.287 |
| Electricity | 96 | 0.126 | 0.218 | 0.128 | 0.221 | 0.375 | 0.437 | 0.129 | 0.224 | 0.131 | 0.229 | 0.140 | 0.237 | 0.132 | 0.229 | 0.129 | 0.222 | 0.148 | 0.240 | 0.193 | 0.308 | 0.201 | 0.317 | 0.164 | 0.269 | 0.168 | 0.272 |
| | 192 | 0.144 | 0.235 | 0.150 | 0.243 | 0.442 | 0.473 | 0.140 | 0.220 | 0.151 | 0.246 | 0.153 | 0.249 | 0.147 | 0.243 | 0.147 | 0.240 | 0.162 | 0.253 | 0.201 | 0.315 | 0.222 | 0.334 | 0.177 | 0.285 | 0.184 | 0.289 |
| | 336 | 0.157 | 0.250 | 0.166 | 0.259 | 0.439 | 0.473 | 0.161 | 0.255 | 0.161 | 0.261 | 0.169 | 0.267 | 0.161 | 0.261 | 0.163 | 0.259 | 0.178 | 0.269 | 0.214 | 0.329 | 0.231 | 0.338 | 0.193 | 0.304 | 0.198 | 0.300 |
| | 720 | 0.183 | 0.276 | 0.185 | 0.276 | 0.980 | 0.814 | 0.194 | 0.287 | 0.197 | 0.293 | 0.203 | 0.301 | 0.196 | 0.294 | 0.197 | 0.290 | 0.225 | 0.317 | 0.246 | 0.355 | 0.254 | 0.361 | 0.212 | 0.321 | 0.220 | 0.320 |
| | Avg | 0.153 | 0.245 | 0.157 | 0.250 | 0.559 | 0.549 | 0.156 | 0.246 | 0.160 | 0.256 | 0.166 | 0.264 | 0.159 | 0.257 | 0.159 | 0.253 | 0.178 | 0.270 | 0.214 | 0.321 | 0.227 | 0.338 | 0.186 | 0.295 | 0.192 | 0.295 |
| Traffic | 96 | 0.357 | 0.236 | 0.358 | 0.242 | 0.843 | 0.453 | 0.360 | 0.249 | 0.376 | 0.264 | 0.410 | 0.282 | 0.336 | 0.253 | 0.360 | 0.249 | 0.395 | 0.268 | 0.587 | 0.366 | 0.613 | 0.388 | 0.519 | 0.309 | 0.593 | 0.321 |
| | 192 | 0.377 | 0.241 | 0.378 | 0.253 | 0.847 | 0.453 | 0.375 | 0.250 | 0.397 | 0.277 | 0.423 | 0.287 | 0.346 | 0.257 | 0.379 | 0.256 | 0.417 | 0.276 | 0.604 | 0.373 | 0.616 | 0.382 | 0.537 | 0.315 | 0.617 | 0.336 |
| | 336 | 0.394 | 0.250 | 0.392 | 0.261 | 0.853 | 0.455 | 0.385 | 0.270 | 0.413 | 0.290 | 0.436 | 0.296 | 0.355 | 0.260 | 0.392 | 0.264 | 0.433 | 0.283 | 0.621 | 0.383 | 0.622 | 0.337 | 0.534 | 0.313 | 0.629 | 0.336 |
| | 720 | 0.439 | 0.283 | 0.434 | 0.287 | 1.500 | 0.805 | 0.430 | 0.281 | 0.444 | 0.306 | 0.466 | 0.315 | 0.386 | 0.273 | 0.432 | 0.286 | 0.467 | 0.302 | 0.626 | 0.382 | 0.660 | 0.408 | 0.577 | 0.325 | 0.640 | 0.350 |
| | Avg | 0.392 | 0.253 | 0.391 | 0.261 | 1.011 | 0.541 | 0.387 | 0.262 | 0.408 | 0.284 | 0.434 | 0.295 | 0.356 | 0.261 | 0.391 | 0.264 | 0.428 | 0.282 | 0.609 | 0.376 | 0.628 | 0.379 | 0.541 | 0.315 | 0.620 | 0.336 |
| Illness (ILI) | 24 | 1.351 | 0.707 | 1.514 | 0.694 | 5.914 | 1.734 | 1.811 | 0.823 | 2.424 | 1.033 | 2.398 | 1.040 | 2.043 | 0.943 | 1.319 | 0.754 | 1.834 | 0.883 | 3.228 | 1.260 | 3.483 | 1.287 | 2.684 | 1.112 | 2.317 | 0.934 |
| | 36 | 1.408 | 0.712 | 1.519 | 0.722 | 6.631 | 1.845 | 1.763 | 0.835 | 2.431 | 1.050 | 2.646 | 1.088 | 1.862 | 0.922 | 1.579 | 0.870 | 1.742 | 0.884 | 2.679 | 1.080 | 3.103 | 1.148 | 2.667 | 1.068 | 1.972 | 0.920 |
| | 48 | 1.434 | 0.721 | 1.500 | 0.725 | 6.736 | 1.857 | 1.705 | 0.818 | 2.459 | 1.061 | 2.614 | 1.086 | 1.796 | 0.920 | 1.553 | 0.815 | 1.996 | 0.938 | 2.622 | 1.078 | 2.669 | 1.085 | 2.558 | 1.052 | 2.238 | 0.940 |
| | 60 | 1.512 | 0.737 | 1.418 | 0.715 | 6.870 | 1.879 | 1.708 | 0.839 | 2.591 | 1.096 | 2.804 | 1.146 | 1.896 | 0.950 | 1.470 | 0.788 | 1.806 | 0.917 | 2.857 | 1.157 | 2.770 | 1.125 | 2.747 | 1.110 | 2.027 | 0.928 |
| | Avg | 1.426 | 0.719 | 1.488 | 0.714 | 6.538 | 1.829 | 1.747 | 0.829 | 2.476 | 1.060 | 2.616 | 1.090 | 1.899 | 0.934 | 1.480 | 0.807 | 1.845 | 0.906 | 2.847 | 1.144 | 3.006 | 1.161 | 2.664 | 1.086 | 2.139 | 0.931 |
| ETTh1 | 96 | 0.359 | 0.386 | 0.368 | 0.395 | 1.044 | 0.773 | 0.361 | 0.390 | 0.361 | 0.392 | 0.375 | 0.399 | 0.375 | 0.398 | 0.370 | 0.400 | 0.386 | 0.405 | 0.376 | 0.419 | 0.449 | 0.459 | 0.421 | 0.431 | 0.384 | 0.402 |
| | 192 | 0.402 | 0.417 | 0.401 | 0.416 | 1.217 | 0.832 | 0.409 | 0.414 | 0.404 | 0.418 | 0.405 | 0.416 | 0.412 | 0.422 | 0.413 | 0.429 | 0.441 | 0.436 | 0.420 | 0.448 | 0.500 | 0.482 | 0.474 | 0.487 | 0.436 | 0.429 |
| | 336 | 0.408 | 0.429 | 0.422 | 0.437 | 1.259 | 0.841 | 0.430 | 0.429 | 0.420 | 0.431 | 0.439 | 0.443 | 0.435 | 0.433 | 0.422 | 0.440 | 0.487 | 0.458 | 0.459 | 0.465 | 0.521 | 0.496 | 0.569 | 0.551 | 0.491 | 0.469 |
| | 720 | 0.419 | 0.448 | 0.441 | 0.465 | 1.271 | 0.838 | 0.445 | 0.460 | 0.463 | 0.472 | 0.472 | 0.490 | 0.454 | 0.465 | 0.447 | 0.468 | 0.503 | 0.491 | 0.506 | 0.507 | 0.514 | 0.512 | 0.770 | 0.672 | 0.521 | 0.500 |
| | Avg | 0.397 | 0.420 | 0.408 | 0.428 | 1.198 | 0.821 | 0.411 | 0.423 | 0.412 | 0.428 | 0.423 | 0.437 | 0.419 | 0.430 | 0.413 | 0.434 | 0.454 | 0.448 | 0.440 | 0.460 | 0.496 | 0.487 | 0.558 | 0.535 | 0.458 | 0.450 |
| ETTh2 | 96 | 0.267 | 0.329 | 0.273 | 0.333 | 2.522 | 1.278 | 0.271 | 0.330 | 0.274 | 0.341 | 0.289 | 0.353 | 0.270 | 0.336 | 0.274 | 0.337 | 0.297 | 0.349 | 0.346 | 0.388 | 0.358 | 0.397 | 0.299 | 0.364 | 0.340 | 0.374 |
| | 192 | 0.338 | 0.375 | 0.340 | 0.378 | 3.312 | 1.384 | 0.317 | 0.402 | 0.339 | 0.385 | 0.383 | 0.418 | 0.332 | 0.380 | 0.314 | 0.380 | 0.400 | 0.429 | 0.439 | 0.456 | 0.452 | 0.486 | 0.454 | 0.454 | 0.402 | 0.414 |
| | 336 | 0.367 | 0.401 | 0.373 | 0.403 | 3.291 | 1.388 | 0.332 | 0.396 | 0.361 | 0.406 | 0.448 | 0.465 | 0.360 | 0.407 | 0.329 | 0.384 | 0.428 | 0.432 | 0.496 | 0.487 | 0.482 | 0.486 | 0.654 | 0.567 | 0.452 | 0.452 |
| | 720 | 0.388 | 0.424 | 0.398 | 0.430 | 3.257 | 1.357 | 0.342 | 0.408 | 0.445 | 0.470 | 0.605 | 0.551 | 0.419 | 0.451 | 0.379 | 0.422 | 0.427 | 0.445 | 0.463 | 0.474 | 0.515 | 0.511 | 0.956 | 0.716 | 0.462 | 0.468 |
| | Avg | 0.340 | 0.382 | 0.346 | 0.386 | 3.095 | 1.352 | 0.316 | 0.384 | 0.355 | 0.401 | 0.431 | 0.447 | 0.345 | 0.394 | 0.324 | 0.381 | 0.383 | 0.407 | 0.433 | 0.447 | 0.453 | 0.462 | 0.588 | 0.525 | 0.414 | 0.427 |
| ETTm1 | 96 | 0.275 | 0.328 | 0.286 | 0.335 | 0.863 | 0.664 | 0.291 | 0.340 | 0.285 | 0.339 | 0.299 | 0.343 | 0.306 | 0.349 | 0.293 | 0.346 | 0.334 | 0.368 | 0.379 | 0.419 | 0.505 | 0.475 | 0.316 | 0.362 | 0.338 | 0.375 |
| | 192 | 0.319 | 0.354 | 0.329 | 0.361 | 1.113 | 0.776 | 0.327 | 0.365 | 0.327 | 0.365 | 0.335 | 0.365 | 0.335 | 0.366 | 0.333 | 0.370 | 0.377 | 0.391 | 0.426 | 0.441 | 0.553 | 0.496 | 0.363 | 0.390 | 0.374 | 0.387 |
| | 336 | 0.353 | 0.374 | 0.358 | 0.379 | 1.267 | 0.832 | 0.360 | 0.381 | 0.356 | 0.382 | 0.369 | 0.386 | 0.364 | 0.384 | 0.369 | 0.392 | 0.426 | 0.420 | 0.445 | 0.459 | 0.621 | 0.537 | 0.408 | 0.426 | 0.410 | 0.411 |
| | 720 | 0.409 | 0.407 | 0.416 | 0.411 | 1.324 | 0.858 | 0.415 | 0.417 | 0.419 | 0.414 | 0.425 | 0.421 | 0.413 | 0.413 | 0.416 | 0.420 | 0.491 | 0.459 | 0.543 | 0.490 | 0.671 | 0.561 | 0.481 | 0.476 | 0.478 | 0.450 |
| | Avg | 0.339 | 0.366 | 0.347 | 0.372 | 1.142 | 0.782 | 0.348 | 0.375 | 0.347 | 0.375 | 0.357 | 0.379 | 0.355 | 0.378 | 0.353 | 0.382 | 0.407 | 0.410 | 0.448 | 0.452 | 0.588 | 0.517 | 0.392 | 0.413 | 0.400 | 0.406 |
| ETTm2 | 96 | 0.157 | 0.244 | 0.164 | 0.250 | 2.041 | 1.073 | 0.164 | 0.254 | 0.163 | 0.252 | 0.167 | 0.260 | 0.161 | 0.251 | 0.166 | 0.256 | 0.180 | 0.264 | 0.203 | 0.287 | 0.255 | 0.339 | 0.179 | 0.275 | 0.187 | 0.267 |
| | 192 | 0.213 | 0.285 | 0.218 | 0.288 | 2.249 | 1.112 | 0.223 | 0.295 | 0.216 | 0.290 | 0.224 | 0.303 | 0.215 | 0.289 | 0.223 | 0.296 | 0.250 | 0.309 | 0.269 | 0.328 | 0.281 | 0.340 | 0.307 | 0.376 | 0.249 | 0.309 |
| | 336 | 0.269 | 0.322 | 0.271 | 0.322 | 2.568 | 1.238 | 0.279 | 0.330 | 0.268 | 0.324 | 0.281 | 0.342 | 0.267 | 0.326 | 0.274 | 0.329 | 0.311 | 0.348 | 0.325 | 0.366 | 0.339 | 0.372 | 0.325 | 0.388 | 0.321 | 0.351 |
| | 720 | 0.351 | 0.377 | 0.361 | 0.380 | 2.720 | 1.287 | 0.359 | 0.383 | 0.420 | 0.422 | 0.397 | 0.421 | 0.352 | 0.383 | 0.362 | 0.385 | 0.412 | 0.407 | 0.421 | 0.415 | 0.422 | 0.419 | 0.502 | 0.490 | 0.408 | 0.403 |
| | Avg | 0.248 | 0.307 | 0.254 | 0.310 | 2.395 | 1.177 | 0.256 | 0.315 | 0.267 | 0.322 | 0.267 | 0.332 | 0.249 | 0.312 | 0.256 | 0.317 | 0.288 | 0.332 | 0.304 | 0.349 | 0.324 | 0.368 | 0.328 | 0.382 | 0.291 | 0.333 |
| Wins | | 20 | 25 | 2 | 4 | 0 | 0 | 3 | 5 | 0 | 0 | 0 | 0 | 5 | 1 | 2 | 1 | 0 | 0 | 0 | 0 | 0 | 0 | 0 | 0 | 0 | 0 |

[a] Taken from Wu et al. (2022).

## A.5   MEASURING THE IMPACT OF VARIATE ORDERING

xLSTM-Mixer fixes one variate ordering to learn multivariate relationships efficiently. We investigate its impact by randomly permuting variate orders and comparing results with the baseline. Tab. 7 shows the results for four such permutations over four horizons on the Weather and ETTm1 datasets, averaged over three initializations each. We observe that the specific ordering does play some role in forecasting performance. However, the standard ordering provided by the dataset sources already permits highly effective forecasting.

Table 7: **Performance of xLSTM-Mixer under variate permutations.**

|  |  | From Dataset | | Perm. #1 | | Perm. #2 | | Perm. #3 | | Perm. #4 | |
|---|---|---|---|---|---|---|---|---|---|---|---|
| Metric | | MSE | MAE | MSE | MAE | MSE | MAE | MSE | MAE | MSE | MAE |
| Weather | 96 | 0.143 | 0.184 | 0.149 | 0.189 | 0.146 | 0.187 | 0.147 | 0.188 | 0.148 | 0.188 |
| | 192 | 0.186 | 0.226 | 0.192 | 0.229 | 0.191 | 0.229 | 0.192 | 0.229 | 0.192 | 0.230 |
| | 336 | 0.236 | 0.266 | 0.242 | 0.269 | 0.241 | 0.269 | 0.242 | 0.269 | 0.240 | 0.269 |
| | 720 | 0.310 | 0.323 | 0.310 | 0.323 | 0.310 | 0.323 | 0.310 | 0.323 | 0.310 | 0.323 |
| ETTm1 | 96 | 0.275 | 0.328 | 0.278 | 0.331 | 0.276 | 0.330 | 0.277 | 0.330 | 0.275 | 0.329 |
| | 192 | 0.319 | 0.354 | 0.321 | 0.356 | 0.321 | 0.356 | 0.320 | 0.355 | 0.319 | 0.355 |
| | 336 | 0.353 | 0.374 | 0.355 | 0.376 | 0.355 | 0.376 | 0.354 | 0.376 | 0.354 | 0.376 |
| | 720 | 0.409 | 0.407 | 0.412 | 0.409 | 0.413 | 0.410 | 0.413 | 0.410 | 0.413 | 0.410 |
| Electricity | 96 | 0.126 | 0.218 | 0.127 | 0.220 | 0.126 | 0.218 | 0.127 | 0.219 | 0.125 | 0.218 |
| | 192 | 0.144 | 0.235 | 0.145 | 0.237 | 0.144 | 0.235 | 0.145 | 0.236 | 0.144 | 0.235 |
| | 336 | 0.157 | 0.250 | 0.160 | 0.252 | 0.159 | 0.251 | 0.157 | 0.248 | 0.159 | 0.250 |
| | 720 | 0.183 | 0.276 | 0.230 | 0.315 | 0.225 | 0.312 | 0.206 | 0.295 | 0.218 | 0.306 |

## A.6   ERROR BARS

This work involves conducting all experiments three times using seeds 2021, 2022, and 2023, following the setup of prior research (Wu et al., 2021; Nie et al., 2023; Wang et al., 2024a). We therefore present the standard deviation of our model and the second-best models in terms of MSE and MAE in Tab. 8. This table, along with our experiments described in Fig. 4 and Fig. 6, further underscores the robustness of xLSTM-Mixer.

Table 8:   The standard deviation for xLSTM-Mixer (ours) and the second-best method in MAE (TimeMixer) and MSE (TiDE), each on ETT, Weather, ILI, Electricity, and Traffic datasets.

| Model | xLSTM-Mixer | | TimeMixer | | TiDE | |
|---|---|---|---|---|---|---|
| Metric | MSE | MAE | MSE | MAE | MSE | MAE |
| Weather | $0.219 \pm 0.000$ | $0.250 \pm 0.000$ | $0.240 \pm 0.010$ | $0.271 \pm 0.009$ | $0.236 \pm 0.001$ | $0.282 \pm 0.001$ |
| Illness | $1.426 \pm 0.046$ | $0.719 \pm 0.011$ | $0.175 \pm 0.045$ | $0.829 \pm 0.120$ | $1.899 \pm 0.001$ | $0.934 \pm 0.004$ |
| Electricity | $0.153 \pm 0.001$ | $0.245 \pm 0.001$ | $0.182 \pm 0.017$ | $0.272 \pm 0.006$ | $0.159 \pm 0.002$ | $0.257 \pm 0.001$ |
| Traffic | $0.392 \pm 0.000$ | $0.253 \pm 0.000$ | $0.484 \pm 0.015$ | $0.297 \pm 0.013$ | $0.356 \pm 0.001$ | $0.261 \pm 0.001$ |
| ETTh1 | $0.397 \pm 0.001$ | $0.420 \pm 0.001$ | $0.047 \pm 0.002$ | $0.440 \pm 0.005$ | $0.419 \pm 0.000$ | $0.430 \pm 0.000$ |
| ETTh2 | $0.340 \pm 0.001$ | $0.382 \pm 0.000$ | $0.364 \pm 0.008$ | $0.375 \pm 0.010$ | $0.345 \pm 0.002$ | $0.394 \pm 0.001$ |
| ETTm1 | $0.339 \pm 0.000$ | $0.366 \pm 0.000$ | $0.381 \pm 0.003$ | $0.395 \pm 0.006$ | $0.355 \pm 0.000$ | $0.378 \pm 0.000$ |
| ETTm2 | $0.248 \pm 0.001$ | $0.307 \pm 0.001$ | $0.275 \pm 0.001$ | $0.323 \pm 0.003$ | $0.249 \pm 0.000$ | $0.312 \pm 0.000$ |

