# OpenReview forum: "xLSTM-Mixer: Multivariate Time Series Forecasting by Mixing via Scalar Memories"
_ICLR.cc/2025/Conference — Submitted to ICLR 2025_

### Official Review · Reviewer_cZwn · 2024-10-27

**Soundness:** 2
**Presentation:** 2
**Contribution:** 1
**Rating:** 3
**Confidence:** 3

**Summary:**

The work proposed xLSTM-Mixer for multivariate time series forecasting that combines linear forecasters (NLinear) with efficient recurrent xLSTM blocks. By adapting xLSTM modules for the temporal and variate dimensions, the model refined the predictions of NLinear with the utilization of time-variate information. The model also employs techniques like reversible instance normalization and soft prompt tokens to further improve the performance. Empirical experiments demonstrate that xLSTM-Mixer can outperform several baseline methods in most cases in long-term forecasting.

**Strengths:**

1. The motivations are well-summarized and the paper's writing is easy to follow.
2. The author provides the code to ensure the reproducibility of the paper.

**Weaknesses:**

1. The main contribution of this paper is to refine the predictions obtained from NLinear by introducing xLSTM modules, which may lack novelty and motivation in technology. On the one hand, please clarify more about the advantages of xLSTM modules that have been refined over (linear forecaster + renorm). For example, to support the mentioned advantages of modeling variate relationships, you can provide a comparative analysis of xLSTM against MLP mixers and efficient attention modules in the Mixer architecture. Also, extending ablation studies (Table 3) to more datasets and different variate-mixing modules would provide a more comprehensive evaluation of the model's effectiveness.
2. It is uncertain whether the authors use the unidirectional xLSTM, which is generally adopted for sequential modeling, to model the correlations of variates. If it is unidirectional, please explain how it addresses the non-sequential nature of the variate correlation, and discuss the potential benefits or drawbacks of using a bidirectional approach in this context.
3. This paper mainly provides the results of long-term forecasting. The lookback seems to be fixed at 96. Considering the extensive overfitting of current deep models on these datasets, it is recommended that the author provide more comparisons with varying lookback lengths and on additional datasets that are less prone to overfitting. This would provide a more robust evaluation of the model's performance.
4. Compared with related recurrent models (such as xLSTMTime), the performance improvement of the proposed model is not significant shown in Table 2. The author should add more recent recurrent models (e.g., SutraNets and P-sLSTM) and more discussions with related work.
5. The showcases in Figure 2 do not reflect the advantages of xLSTM-Mixer in multivariate modeling. To make it more specific, please include visualizations that explicitly show how xLSTM-Mixer captures relationships between multiple variates in its predictions.
6. The paper concludes by highlighting the potential of xLSTM-Mixer for future research in areas such as short-term forecasting, time series classification, and imputation. I wonder if there are some results on these analysis tasks using xLSTMs.

**Questions:**

1. Has the author tried to train on time series with randomly shuffled temporal orders, and what will happen to the results?
2. About the experiments of enlarging lookback (Figure 3): In the DLinear paper, the performance can also be improved by increasing the lookback using a single linear layer. It is suggested that the author introduce such linear models into the baseline. Otherwise, the advantage can only be due to NLinear. It is also encouraged to provide the model performance with a longer lookback length to verify whether the proposed can continuously improve the performance (or keep it stable in a very long lookback length).

---

> ### Author Response · Authors · 2024-11-20
>
> Thank you for your detailed review of our paper and the time you took to evaluate our work, as well as for highlighting opportunities for improvement. We are pleased that you found the motivations well-summarized and the writing easy to follow. Below, we address each of the points you raised in detail:
>
> - **W1 (key properties of xLSTM-Mixer)**: We are happy to elaborate further. The key contribution of xLSTM-Mixer lies in the careful integration of the three mixing steps and a thorough evaluation of their effectiveness. As shown in Table 3, the use of xLSTM modules alone, without additional projections, normalizations, or appropriate weight-sharing, results in poor performance. Identifying the effective combination for forecasting and motivating the choices in Sections 2.2 and 3 are significant advancements. We have now extended the ablation study to include an aggregate of 288 experiments. Specifically, as suggested, we investigated the refinement using mLSTM instead of sLSTM blocks (column #2) and different normalization strategies (column #4).
>
> - **W2 (bidirectional xLSTM)**: Thank you for pointing this out. We employed unidirectional xLSTM models in our work, which indeed fixes the order of variates. Despite this limitation, combined with the advancements discussed in W1, our approach still achieved state-of-the-art performance. Extending to bidirectional xLSTMs is a promising direction for future work, as we outlined (cf. lines 543ff). Please also refer to G3 for more insights regarding the  order of variates.
>
> - **W3 (lookback length)**: The lookback length was a hyperparameter selected through a search process, as outlined in Appendix A.1/Table 4. In line with best practices in forecasting evaluation, we conducted out-of-sample testing by splitting off the test portion from the end of the timeline (using the same data splits as previous works [1,2,3,4]), effectively addressing overfitting concerns. We agree that exploring different lookback lengths is crucial for sensitivity analysis, and we have already investigated this in Figure 6 (see lines 479-488). We have now extended the lookback analysis to include lengths up to 2048 and discuss this in the revised manuscript.
>
> - **W4 (more recent work)**: We respectfully disagree regarding the significance of the improvement, as xLSTM-Mixer outperforms xLSTMTime with 45 wins compared to 6 (see updated Table 2), often by a very large margin. However, we agree that recent recurrent models should be discussed more in the related work section, and we have updated that section to reference the mentioned methods. We chose not to include them in Table 2 since models like P-sLSTM would not achieve even a second-place ranking in our benchmark across datasets.
>
> - **W5 (capturing multivariate relationships)**: Thank you for this interesting suggestion. We have conducted additional experiments to highlight how xLSTM-Mixer captures multivariate relationships. Please refer to G2.
>
> - **W6 (outlook)**: The focus of this work is long-term forecasting, following recent influential publications such as PatchTST (Nie et al. [2]), MICN (Wang et al. [5]), TiDE (Das et al. [7]), and N-HiTS (Challu et al., [6]). We provided initial short-term forecasting experiments in Appendix A.2. We defer further exploration of tasks such as classification, imputation, or anomaly detection to future work. The strong performance in long-term forecasting suggests promising future research opportunities.
>
> - **Q1 (temporal ordering)**: This is an interesting idea; we thought about it too. The main change would likely be a deterioration of the normalization in the initial forecast, which relies on the magnitude of the last time step being representative of the subsequent forecast horizon. However, the succeeding components, namely starting with $\operatorname{FC}^\text{up}(\cdot)$, would not be affected by the dataset being shuffled that way. This is because the fully-connected layer can model connections of any hidden state to any time step. Thereby, the hidden representations are not bound to fixed time steps and can model arbitrary relationships between any number of timesteps in a flexible manner.
>
> - **Q2 (lookback length and linear baselines)**: Adding LTSF baselines from Zeng et al. [4] is a great suggestion to further strengthen our claims. Consequently, we have updated Figure 5 to also include NLinear.
>
> Thank you once again for your valuable feedback and constructive suggestions. We are confident that these changes have strengthened the manuscript and are happy to provide any further clarifications you may need.

---

> > ### Author Response · Authors · 2024-11-20
> > **References**
> >
> > [1] Wang, S., et al. (2024). TimeMixer: Decomposable multiscale mixing for time series forecasting. ICLR
> >
> > [2] Nie, Y., et al. (2023). A time series is worth 64 words: Long-term forecasting with transformers. ICLR
> >
> > [3] Wu, H., et al. (2021). Autoformer: Decomposition transformers with auto-correlation for long-term series forecasting. NeurIPS
> >
> > [4] Zeng, A., et al. (2023). Are transformers effective for time series forecasting? AAAI
> >
> > [5] Wang, H., et al. (2023). MICN: Multi-scale local and global context modeling for long-term series forecasting. ICLR
> >
> > [6] Challu, C., et al. (2022). N-HiTS: Neural Hierarchical Interpolation for Time Series Forecasting. ArXiv
> >
> > [7] Das, A., et al. (2023). Long-term forecasting with TiDE: Time-series dense encoder. TMLR

---

> > > ### Author Response · Authors · 2024-11-21
> > > **Any further questions?**
> > >
> > > Dear Reviewer,
> > >
> > > We hope to have resolved all your concerns. If there are any further comments from your side, we will be happy to address them before the rebuttal period ends. If there are none, then we would appreciate it if you could reconsider your rating.
> > >
> > > Regards,
> > >
> > > Authors

---

> > > > ### Comment · Reviewer_cZwn · 2024-11-25
> > > >
> > > > Thank you for the revised paper and replies to my review. I have read the contents of the rebuttal and the revision carefully. But here are a few comments based on the new version:
> > > >
> > > > 1. Regarding W1: Despite further experimentation provided by the authors, the conclusion comes to weaken the contribution of using xLSTM since the use of normalization or weight-sharing on these benchmarks would have improved the results themselves. However, I have not seen specific bonding to the proposed architecture.
> > > >
> > > > 2. Regarding W2: Appendix A.5 provides the forecast performance on the shuffled variates, and the result shows that there is nearly no difference in performance, which further confirms that it is unnecessary to distinguish the order of variables. Therefore, I think there is no support for using directional structures (such as unidirectional xLTSM) for variables.
> > > > 3. Regarding W3: the search of the lookback window leads to an unfair comparison of several baseline models, I find that the several baseline results currently are reported under a fixed-96 forecast length.
> > > >
> > > > Given the current performance and the innovation of this paper, I'd like to maintain my current score.

---

> ### Author Response · Authors · 2024-11-25
>
> Thank you once again.
>
> - **W1** We believe this showcases the need for an initial forecast, but it is task-dependent whether we use NLinear or DLinear. This further emphasizes the importance of the xLSTM refinement along with our other contributions. Without these contributions, we would be limited to the results of either NLinear or DLinear. Moreover, our contribution was not only to evaluate whether normalization is necessary; we also conducted extensive studies, such as using initial token embeddings, exploring xLSTM block variants, analyzing explainability, and comparing the influence of time and variates similar to iTransformer. These are shown in our ablation study in Table 3, many variants of simply using xLSTM or NLinear/DLinear do not yield SOTA results. Therefore, we conducted an extensive evaluation of possible architectures and provided a detailed analysis in Section 4.2 to thoroughly assess the properties of the resulting model. Lastly, our model has demonstrated SOTA performance while requiring very limited hardware resources.
> Therefore, we argue against the notion that our contribution is merely incremental.
>
> - **W2** Thank you for that comment. We conducted another experiment using a dataset with significantly more variates (see the updated table 7) and found that, for longer forecasting horizons, there is indeed a significant difference in the ordering. The initial ordering appears to provide results according to the chosen hyperparameters. This supports the rationale for using a unidirectional xLSTM. However, as seen in datasets with fewer variates and lower forecasting horizons, we believe that employing a bidirectional (x)LSTM can further improve the already SOTA performance. This is why we are excited to see this explored in future work.
>
> - **W3** We assume you probably mean a fixed lookback length of 96, since all baselines are evaluated across four forecast horizons, just as we have done. We agree that comparability is key in this study, and therefore, we double-checked to ensure the results are directly comparable. For example, previous SOTA methods such as TimeMixer (Wang et al., 2024, see specifically Table 14 on p. 18), TiDE (Das et al., 2024, see specifically Section 5.1, "Baselines and Setup", "the look-back window was tuned in {24, 48, 96, 192, 336, 720}"), and NLinear/DLinear (Zeng et al., 2023, Table 2) all compare under "optimal" lookbacks when reporting their results. In practical applications, one would nearly always optimize the lookback length. We acknowledge that some works compare under a fixed lookback length, which is unfortunately not always set to 96 (e.g., Autoformer uses a different value, such as 512, as in xLSTMTime). Thus, we are confident that our finding that xLSTM-Mixer provides superior forecasting accuracy is reliable.
>
> We hope we have convinced you that our contribution is non-trivial as well as the new experiments have sufficiently answered your outstanding concerns. We will be happy to answer any further questions and hope that the reviewer reconsiders the score.

---

> > ### Author Response · Authors · 2024-11-27
> > **Any further questions?**
> >
> > Dear Reviewer,
> >
> > We hope to have clarified your outstanding concerns. Let us know if there are any further questions. If no, it will be great if you could reconsider your score.
> >
> > Regards,
> >
> > The Authors

---

> ### Author Response · Authors · 2024-12-02
>
> Dear Reviewer cZwn,
>
> Thank you for the constructive discussion. We hope to have clarified all remaining questions.
>
> As the reviewing period will soon close, we would kindly ask you to reconsider your currently unchanged score after our extensive clarifications and significant additions. We specifically addressed your constructive comments in the revised manuscript and comments.
>
> Regards,
>
> The Authors

---

### Official Review · Reviewer_FwwC · 2024-10-30

**Soundness:** 4
**Presentation:** 3
**Contribution:** 4
**Rating:** 8
**Confidence:** 4

**Summary:**

The paper introduce  a novel time series forecasting model with time and variate mixing in the context of recurrent models.

**Strengths:**

- Figure 1 provides a clear overview of the method.
- Several commonly used benchmark datasets are included.
- The proposed xLSTM-Mixer model consistently achieves state-of-the-art performance across a broad range of benchmarks.
- Informative forecast plots are included.
- A comprehensive set of baseline models from different architecture categories including, Transformer-based, CNN-based, MLP-based, and Recurrent architectures.
- The ablation study on architecture choices is highly insightful and supports the combined aspects of the proposed model.
- Open-source code is available with reproducible experiments

**Weaknesses:**

1. Random seeds were included as a hyperparameter, but it would be beneficial to train the model over several trials to compute the average and standard deviation of the results. This approach would provide a more robust measure of performance consistency across runs.
2. The approach does not reference similar architectures, such as iTransformer, which utilizes the attention and feed-forward network on inverted dimensions. In iTransformer, individual time points are embedded as variate tokens, leveraging the attention mechanism to capture multivariate correlations. This bears a resemblance to the proposed method, which jointly combines time and variate information to capture complex patterns in the data by employing transposed input dimensions to invert the series. Mentioning this may help emphasize the translatable similarities across architecture backbones.
3. The motivation for the reverse-up projected embedding is not explicitly explained. Including an ablation study of the model with and without this embedding would be useful to help justify this architectural choice.
4. Some models cited in prior work are missing as baselines, including TFT, N-HiTS, and N-BEATS.

**Questions:**

Suggestions:
- Adding dimensions to Fig. 1 would help clarify input shapes after higher-dimensional projections and transpositions.
- Lines 41-42: Regarding Transformer architectures, the authors state, “For instance, they typically require large datasets to train successfully, restricting their use to only a subset of applications.” However, I do not believe this claim is fully supported by the literature. For example, common benchmarks for long-horizon forecasting include datasets like ILI, Traffic, and Weather. The ILI dataset, with only 966 timestamps, could be considered relatively small, yet PatchTST outperforms other Transformer and linear models on this dataset [1]. Similarly, for ILI, Autoformer outperforms other models, including recurrent architectures like LSTMs [2].
- It would help motivate the proposed approach to illustrate the trade-offs in computational complexity between the proposed model, Transformer-based models, and MLP-based models for both training and inference.
- While model hyperparameters for the proposed model are included in the appendix, it would be helpful to provide hyperparameters and the number of training epochs for the baseline models as well.
- Additional benchmark datasets could include ILI and Exchange datasets which are used in prior work [2].

References:
1. “A Time Series is Worth 64 Words: Long-term Forecasting with Transformers”
2. “Autoformer: Decomposition Transformers with Auto-Correlation for Long-Term Series Forecasting”

---

> ### Author Response · Authors · 2024-11-20
>
> Thank you very much for your positive and insightful review of our paper. We are delighted that you found our work well-presented and appreciated our contributions, including the thorough ablation study. We value your suggestions for improvement, and below we provide detailed responses to each of your comments.
>
> - **W1 (seeds)**: We fully agree that performing experiments with multiple seeds is crucial for obtaining reliable results. Therefore, all experiments, including those in the appendix, were conducted with three different seeds, and the averages were reported. This information is available in the technical details in Appendix A.1, under the "Training and Hyperparameters" section.
>
> - **W2 (relation to iTransformer)**: You are correct in noting the similarity to the iTransformer by Liu et al. [1]. We have now referenced this work in lines 206-208 and in Section 2.3. We believe these references are helpful in understanding the context, but we are open to extending the discussion further if you think it is necessary.
>
> - **W3 (motivation for the reverse-up projection)**: We motivate the multi-view mixing and $\mathbf{\widehat x}^\text{up}$ in Sec. 3.3. Indeed, the ablation is super interesting. Therefore, we did that in Tab. 3., where "Mix View" indicates if that exact multi-view mixing with the reversed up-projection has been included or if only a single xLSTM pass was performed.
>
> - **W4 (reasoning for the choice of baselines)**: Thank you for pointing out the omission of certain baselines. Due to space constraints, we opted to focus on a subset of recent baselines that provide challenging benchmarks. Specifically, we replaced older baselines (e.g., TFT, N-BEATS, and N-HiTS) with more recent models that are significantly better across multiple categories—Transformers, MLPs, and others. Including these newer, stronger baselines ensures a more competitive evaluation.
>
> - **Q1 (adding dimensions to Fig. 1)**: We appreciate this suggestion. While we found that adding dimensions to Figure 1 might have made it too cluttered, we instead added more descriptive information about intermediate dimensions in Sections 3.1 to 3.3 to assist readers in understanding the architecture more clearly.
>
> - **Q2 (data requirements for transformers)**: Thank you for double-checking these claims. We were inspired by language modeling, where multiple studies suggest that model performance scales with increasing dataset size/token counts [2,3]. However, we fully agree with your assessment regarding time-series models. We removed this claim from ll. 41f in the revised PDF.
>
> - **Q3.1 (computational efficiency)**: We appreciate this suggestion for a more in-depth analysis. We have now conducted a thorough benchmarking of xLSTM-Mixer against key baselines, including Transformer-based and MLP-based models, for both training and inference. Please refer to G3.
>
> - **Q3.2 (hyperparameters)**: For the hyperparameters of the baseline models, we primarily relied on the Time-Series-Library [4]. For models not covered there, such as LSTM and xLSTMTime, we used information from [5] and [6], respectively.
>
> - **Q4 (further datasets)**: This is indeed a useful extension. We chose to include Illness (*ILI*) as the more challenging of the two datasets in the extended Table 2. We extended the dataset overview in Table 1 to include the Hurst exponent to better judge the complexity of the dataset. Lower numbers entail more erratic long-term behavior than higher ones. Below, you can find the full table, including Exchange:
>
> | Dataset     | Domain           | Horizons    | Sampling     | #Variates | Hurst exp.        |
> |-------------|------------------|-------------|--------------|-----------|-------------------|
> | Weather     | Weather          | 96--720     | 10 min       | 21        | 0.333--1.000      |
> | Electricity | Power Usage      | 96--720     | 1 hour       | 321       | 0.555--1.000      |
> | Traffic     | Traffic Load     | 96--720     | 1 hour       | 862       | 0.162--1.000      |
> | ETT         | Power Production | 96--720     | 15&60 min    | 7         | 0.906--1.000      |
> | Illness (ILI) | Influenza cases | 24--60      | 1 week       | 7         | 0.499--0.907      |
> | Exchange    | Stocks           | 96--720     | 1 day        | 8         | 0.968--1.000      |
>
> Thank you once again for your valuable comments. We are happy to answer any further questions or provide additional clarifications if needed.

---

> > ### Author Response · Authors · 2024-11-20
> > **References**
> >
> > [1] Liu, Y. (2024). iTransformer: Inverted Transformers are effective for time series forecasting. ICLR
> >
> > [2] Hoffmann, J., et al. (2022). An empirical analysis of compute-optimal large language model training. NeurIPS
> >
> > [3] Kaplan, J., et al. (2020). Scaling laws for neural language models. ArXiv
> >
> > [4] Wang, Y. (2024). Deep time series models: A comprehensive survey and benchmark. ArXiv
> >
> > [5] Alharthi, M., & Mahmood, A. (2024). xLSTMTime: Long-Term Time Series Forecasting with xLSTM. AI, 5(3), 1482-1495.
> >
> > [6] Wang, H., et al. (2023). MICN: Multi-scale local and global context modeling for long-term series forecasting. ICLR

---

> > > ### Author Response · Authors · 2024-11-21
> > > **Any further questions?**
> > >
> > > Dear Reviewer,
> > >
> > > We hope to have resolved all your concerns. If there are any further comments from your side, we will be happy to address them before the rebuttal period ends.
> > >
> > > Regards,
> > >
> > > Authors

---

> > > > ### Comment · Reviewer_FwwC · 2024-11-24
> > > >
> > > > Thank you for addressing my questions and concerns. Regarding W1, including the information that the results are reported as averages over multiple random seeds in the main paper would be relevant. Additionally, I strongly recommend including the standard deviations in the appendix, as this would provide a clearer understanding of the variance across runs and help establish whether significant performance margins exist between models.

---

> > > > > ### Author Response · Authors · 2024-11-24
> > > > > **New Table in A.6 + Revision**
> > > > >
> > > > > Thank you once again for your valuable feedback and suggestions to improve our manuscript.
> > > > >
> > > > > We have revised the main text to incorporate your recommendation to specify that we use multiple seeds. Additionally, we have included a new table with error bars in the appendix (refer to A6), showcasing the three best-performing methods (including ours) based on their wins according to MSE and MAE in our benchmarks.
> > > > >
> > > > > Your insights have helped to improve our manuscript even further. Thank you again for your thoughtful review!
> > > > > Please don't hesitate to ask if you have any remaining comments.

---

### Official Review · Reviewer_9duf · 2024-11-01

**Soundness:** 2
**Presentation:** 3
**Contribution:** 3
**Rating:** 5
**Confidence:** 4

**Summary:**

The paper presents a novel design - xLSTM-Mixer which is based on LSTM and a set of mixing machanism for multivariate time series forecasting. The model builds on the idea of recurrent structures, integrating three mixing strategy: an initial linear forecast with time mixing, joint mixing with sLSTM blocks, and a final view mixing stage. Evaluations show that xLSTM-Mixer achieves state-of-the-art forecasting performance across multiple datasets which is a key contribution to the ongoing development of recurrent models.

**Strengths:**

**Innovation in Model Architecture**: The integration of time and variate mixing with xLSTM blocks is an innovative approach. This design tackles some limitations of conventional recurrent and transformer-based models in capturing complex temporal dependencies, which is especially challenging in multivariate time series data.

**Empirical Performance**: The experiments show that xLSTM-Mixer outperforms a variety of established methods (e.g., Time-Mixer, iTransformer, MICN) in terms of mean squared error (MSE) and mean absolute error (MAE) across multiple datasets, together with comprehensive ablation studies on initializations, size of hidden dimension and length of lookback window shows the model’s robustness and generalizability.

**Weaknesses:**

**Model structure**:
LSTM-Mixer appears to be an incremental improvement over xLSTM, and the mixing machisim does not seems to be very siginificant regards to the Ablation Study on Table 3. An experiment with alternative recurrent structures may help here. Instead of relying solely on the sLSTM block with the joint mixing machinism, it would be usefule to assess the performance of other recurrent models, such as GRU and Mamba.

**Computational Efficiency**:
The complexity of xLSTM-Mixer’s mixing machanism and recurrent structures would be computationally intensive. Further discussion on the computational costs relative to attentions and SSMs would be beneficial.

**Evaluation on Short-Term Forecasting**:
While the model demonstrates strong performance in long-term forecasting, the short-term forecasting evaluation is inadequate. Adding a more robust short-term evaluation could highlight the model’s flexibility across different forecasting horizons.

**Evaluation on Long-Term Forecasting**:
Although most studies follow the 96, 192, 336, 720 prediction length on benchmarks, regards to this work highlighted long-term forecasting as a distinctive feature, a prediction length of 720 is insufficient.

**Questions:**

**More discussion on the use of Multivariate information**:
- Firstly, the primary innovation claimed by the authors is the use of multivariate time series forecasting. However, simply explaining how the new method works and benchmarking with other multivariate models is insufficient. Since the impact of using multivariate information varies across different datasets and variables, and given that multivariate mixing is the core innovation, additional analysis on this aspect should be provided.
- Secondly, whats are the authors' opinion on the heterogeneity of variates, and how important is this compare to the temporal dependency in terms of forecasting?
- Thirdly, the authors mention that variate ordering may impact performance but leave this for future work. Given the importance of variate dependencies in time series, a preliminary exploration of this factor could provide additional insights into potential model improvements.

---

> ### Author Response · Authors · 2024-11-20
>
> We appreciate the reviewer’s thoughtful review and recognition of the key strengths of our work, particularly the acknowledgment of our innovative model architecture and empirical performance of xLSTM-Mixer. In the following we aim to address the concerns raised.
>
> - **W1 (novelty and significance)**: You are right in that view mixing provides only a small benefit on Weather. However, comparing #1 and #6 in (the updated) Tab. 3 on ETTm1 shows a rather consistent improvement due to Time Mixing. We agree that more ablations with other recurrent models would be helpful, and therefore replaced the sLSTM blocks with mLSTM blocks (see the new column #2). GRUs were not originally built for multivariate data, for which reason we opted for mLSTM. We furthermore show that even the selection of initial decomposition is non-trivial (new column #4). In summary, it shows that xLSTM-Mixer is a carefully crafted model with significant efforts beyond incremental changes required for its effectiveness.
>
> - **W2 (study on computational efficiency)**: We totally agree that this would help better estimate the practical benefits of xLSTM-Mixer in relation to previous work. Therefore, we added an additional study on run time and peak memory usage, making sure to compare to models previously presented in Sec. 4.1. Please see G3 for details.
>
> - **W3 (focus on long-term forecasting)**: We focus on effective long-term forecasting, as has also been done by the famous works PatchTST (Nie et al. [2]), MICN (top 5% ICLR; Wang et al. [3]) and many other, including the renowned TiDE (Das et al., [5]) and N-HiTS (Challu et al. [4]). To still embark on further challenges as guidance for future work, we provide the short-term as an outlook. It is placed in the Appendix as it is not the core focus of the paper. We see this as sufficient and encourage readers to investigate further. Furthermore, while xLSTM-Mixer overall places second on short-term evaluation after TimeMixer, the study on computational efficiency showed that this comes with the benefit of greatly reduced memory requirements.
>
> - **W4 (more horizons)**: We aim to be maximally comparable to the previous literature. This is facilitated by meticulously replicated setups and best practices, including the predictions horizons also used by xLSTMTime (Alharthi et al. [6]), TimeMixer (Wang et al. [1]), TSMixer (Chen et al. [8]), NLinear/DLinear (Zeng et al. [7]), iTransformer (Liu et al. [9]), FEDFormer (Zhou et al. [10]), Autoformer (Wu et al. [11]), MICN (Wang et al. [3]), TimesNet (Wu et al. [12]), and many more, including ones that exclusively perform forecasting, such as TiDE (Das et al. [5]), N-BEATS (Oreshkin et al. [13]), N-HiTS (Challu et al. [4]), PatchTST (Nie et al. [2]). We agree with these previous works, which, based on an investigation of the respective origins of datasets, arrived at the commonly used selection of forecast horizons.
>
> - **Q1 (analyzing the learned cross-variate relationships)**: We agree that investigating this would shed further light on the efficacy of xLSTM-Mixer. We thus performed an attribution study that we provide in our revised mansucript. Please refer to G2 for details.
>
> - **Q2 (variate heterogeneity and temporal patterns)**: The temporal patterns are key to (long-term) forecasting, as demonstrated by the effectiveness of channel-independent architectures such as NLinear/DLinear by Zeng et al. [7]. However, as shown by our variate attribution study in the revised manuscript (lines 409-419 and Fig. 5), capturing relationships between variates further enhances forecast accuracy. Empirically, the heterogeneity of the variates in the considered datasets does not pose a challenge to xLSTM-Mixer, presumably due to the highly expressive sLSTM blocks.
>
> - **Q3 (variate ordering)**: We completely agree that such a study would greatly strengthen the evaluation and, therefore, conducted the necessary experiments. Please see G2 for the respective findings.
>
> Thank you once again for your valuable comments. We are happy to answer any further questions or provide additional clarifications if needed.

---

> > ### Author Response · Authors · 2024-11-20
> > **References**
> >
> > [1] Wang, S., et al. (2024). TimeMixer: Decomposable multiscale mixing for time series forecasting. ICLR
> >
> > [2] Nie, Y., et al. (2023). A time series is worth 64 words: Long-term forecasting with transformers. ICLR
> >
> > [3] Wang, H., et al. (2023). MICN: Multi-scale local and global context modeling for long-term series forecasting. ICLR
> >
> > [4] Challu, C., et al. (2022). N-HiTS: Neural Hierarchical Interpolation for Time Series Forecasting. ArXiv
> >
> > [5] Das, A., et al. (2023). Long-term forecasting with TiDE: Time-series dense encoder. TMLR
> >
> > [6] Alharthi, M. and Ausif M. (2024): xLSTMTime: Long-Term Time Series Forecasting with xLSTM. MDPI AI
> >
> > [7] Zeng, A., et al. (2023). Are transformers effective for time series forecasting? AAAI
> >
> > [8] Chen, S.-A., et al. (2023). TSMixer: An all-MLP architecture for time series forecasting. TMLR
> >
> > [9] Liu, Y., et al. (2024). iTransformer: Inverted Transformers are effective for time series forecasting. ICLR
> >
> > [10] Zhou, H., et al. (2022). FEDformer: Frequency Enhanced Decomposed Transformer for Long-term Series Forecasting. ICML
> >
> > [11] Wu, H., et al. (2021). Autoformer: Decomposition Transformers with Auto-Correlation for Long-Term Series Forecasting. NeurIPS
> >
> > [12] Wu, H., et al. (2022). TimesNet: Temporal 2D-Variation Modeling for General Time Series Analysis. ICLR
> >
> > [13] Oreshkin, B. N., et al. (2019). N-BEATS: Neural basis expansion analysis for interpretable time series forecasting. ICLR

---

> > > ### Author Response · Authors · 2024-11-21
> > > **Any further questions?**
> > >
> > > Dear Reviewer,
> > >
> > > We hope to have resolved all your concerns. If there are any further comments from your side, we will be happy to address them before the rebuttal period ends. If there are none, then we would appreciate it if you could reconsider your rating.
> > >
> > > Regards,
> > >
> > > Authors

---

> ### Comment · Reviewer_9duf · 2024-11-25
>
> Thank you for the response. The authors provided additional statement and analyses adds credibility to the novelty claim, the Comprehensive Evaluation of Computational Efficiency confirms that xLSTM-Mixer is computationally efficient relative to comparable models regards to the aspect of time versus model size. However, despite the improvements and ablation studies, the overall design of xLSTM-Mixer still leans towards incremental innovation, the added experiments with mLSTM blocks improve this aspect but do not fully establish paradigm-shifting advancements. The additional analysis and further experiment on learning multivariate information variate ordering does not really show a significant impact on the performance, the experiments remain relatively preliminary and may not be sufficient to fully address the broader impact of variable ordering across different datasets and tasks. Although the new experiments and materials provided by the authors are limited in depth, they go beyond the initially stated “future work” scope, demonstrating the model’s potential in this area. Therefore, this effort is reasonable and commendable. I will therefore raise my score.

---

### Official Review · Reviewer_GoVE · 2024-11-02

**Soundness:** 3
**Presentation:** 2
**Contribution:** 2
**Rating:** 3
**Confidence:** 4

**Summary:**

This paper introduces a new time series prediction model called xLSTM-Mixer, which aims to effectively capture and integrate complex patterns in time series data by combining linear prediction with xLSTM blocks. The model first uses a shared linear model to make initial predictions for each variable, then fine-tunes and mixes these predictions through a series of sLSTM modules, and finally generates the final prediction result by integrating two different perspectives' predictions. Experimental results show that xLSTM-Mixer has significant advantages in long-term time series prediction tasks, achieving better predictive accuracy than current state-of-the-art methods on multiple benchmark datasets. In addition, the article conducts detailed analysis of the model, including A/B testing, sensitivity analysis of hyperparameters, and research on initial embedding vectors, further validating the effectiveness and robustness of the model. This work not only improves the level of deep learning-based time series prediction technology but also provides new directions for future related research.

**Strengths:**

- Superior long-term forecasting performance: xLSTM-Mixer outperforms recent state-of-the-art methods in terms of predictive accuracy on various benchmark datasets.

- Robustness and effectiveness: Detailed model analysis, including A/B testing, sensitivity analysis of hyperparameters, and research on initial embedding vectors, further validates the effectiveness and robustness of the model.

- Contribution to the resurgence of recurrent models in time series forecasting: By combining linear prediction with xLSTM blocks and integrating time, variate, and multi-view mixing, xLSTM-Mixer enhances the learning of necessary features and offers a viable alternative to current sequence models, contributing to the resurgence of recurrent models in time series forecasting.

**Weaknesses:**

- The contribution regarding advancement of  xLSTM-Mixer over the base xLSTM model is not clearly defined. The novelty and significant contributions need further clarification to distinguish xLSTM-Mixer as more than a trivial modification.

- The paper asserts that longer lookback lengths improve forecasting but Figure 5 shows increased MSE at a lookback length of 1024 compared to 768. This requires further exploration as it conflicts with the claims.

- The manuscript dismisses trend-seasonality decomposition without empirical support, which is a significant weakness given the technique's established value in forecasting.

- The paper omits a comparison of the xLSTM-Mixer's computational complexity and cost against other methods, a critical evaluation factor for practical model deployment.

**Questions:**

- The manuscript fails to convincingly argue the substantial novelty of xLSTM-Mixer compared to prior xLSTM models. Authors should clarify that xLSTM-Mixer is not merely a combination of xLSTM and Multi-View Mixing, as the current description lacks the depth to establish its significance.

- While the manuscript acknowledges xLSTM-Mixer's performance with varying lookback windows (Figure 5), it lacks a thorough experimental setup to explore different input/output ratios. A comprehensive analysis should include varied input lengths relative to output horizons to assess model robustness. The absence of this setup limits understanding of performance in diverse forecasting conditions. Additional experiments with varying input/output ratios, as suggested by Table 6, are recommended.

- The paper does not explain the increase in MSE at a lookback length of 1024, leaving readers unclear about the model's behavior. To strengthen the manuscript, the authors should provide a plausible explanation or conduct additional experiments to verify performance consistency across different lookback lengths.

- A significant issue is the claim that a simple NLinear normalization scheme is sufficient without rigorous testing or theoretical backing. The decision to forgo trend-seasonality decomposition and rely solely on NLinear normalization lacks the necessary validation, raising questions about the model's robustness. More experimental corroboration against established methods is needed.

---

> ### Author Response · Authors · 2024-11-20
>
> Thank you for your detailed review of our paper. We appreciate the time and effort you put into evaluating our work and your constructive feedback. Your comments are invaluable in improving the manuscript even further, and we are pleased that you recognized the superior long-term forecasting performance, robustness of our model, and our contributions to recurrent model research. Below, we address the opportunities for improvement and questions you raised.
>
> - **W1/Q1 (contribution)**: We argue that xLSTM is not just a "base model," but a new framework and family of models [1], similar to Convolutional Neural Networks and Transformers, which have now been adapted for various tasks, including vision, robotics, and other domains [2,3,4,5]. Our work enhances their effectiveness specifically for long-term time series forecasting. A straightforward application of xLSTMs to time series data is insufficient, as demonstrated by the variations examined in our ablation study (Table 3). We contribute several essential components, including normalization, a careful combination of time and variate mixing, key up- and down-projections, and the introduction of initial token embeddings. Furthermore, view mixing is one of our original contributions. We also provide extensive analysis, including comparisons to 12 baselines across seven datasets (now eight datasets, see G2), forecast visualizations, a comprehensive ablation study, sensitivity analysis of hidden dimensions and lookbacks, and visualizations of the learned initial tokens. Following the rebuttal, we are the first to explore feature importance for xLSTMs in forecasting and, to the best of our knowledge, also for general xLSTMs.
>
> - **Q2 (explore different input/output ratios)**: Figure 5 already includes all possible combinations of the lookback steps ($T$) and forecast horizons ($H$) for each model. Are you suggesting we present these results differently? We would be happy to revise the presentation if it would improve clarity. See also the next response for further details.
>
> - **W2/Q3 (lookback window in Fig. 5)**: Thank you for pointing this out. We extended the lookback length of our model and most baselines across all four scenarios to also include $T=2048$, updating Figure 5 accordingly. This addition helps illustrate the underlying dynamics more clearly:
>     - For short prediction lengths, using more than 768 time steps in the past becomes redundant, slightly impairing the model's performance.
>     - For longer horizons, extending the lookback becomes increasingly beneficial for forecasting.
>     - xLSTM-Mixer demonstrates consistently strong and robust performances across all scales.
>
> - **W3/Q4 (trend-seasonality)**: Thank you for suggesting more empirical support for our choice to forgo trend-seasonality decomposition. We have now included a thorough ablation study to evaluate this decision further. Please refer to G1 for additional insights.
>
> - **W4 (computational complexity)**: Thank you for pointing this out. We agree that evaluating xLSTM-Mixer's practical deployment capabilities requires a comparison of computational complexity and cost against other methods. In addition to the extended experiments involving very long lookbacks (as reflected in the updated Figure 5), which demonstrate xLSTM-Mixer's practical applicability, we also quantified its runtime and memory requirements relative to the baselines. See G3 for details. Our findings show that xLSTM-Mixer is well-suited for devices with limited resources compared to other methods (requiring in some occasions ~160 times less GPU memory than PatchTST while maintaining similar speed) and still achieves SOTA performance. We appreciate the suggestion to conduct this important experiment, which has significantly improved the manuscript. In summary, xLSTM-Mixer is lightweight compared to other recent architectures, including PatchTST, in both memory requirements and compute time, while still providing SOTA results.
>
> Thank you once again for your valuable comments. We are happy to answer any further questions or provide additional clarifications if needed.

---

> > ### Author Response · Authors · 2024-11-20
> > **References**
> >
> > [1] Beck, M., et al. (2024). xLSTM: Extended long short-term memory. ArXiv
> >
> > [2] Alkin, B., et al. (2024). Vision-LSTM: xLSTM as generic vision backbone. ArXiv
> >
> > [3] Schmidinger, N., et al. (2024). Bio-xLSTM: Generative modeling, representation and in-context learning of biological and chemical sequences. ArXiv
> >
> > [4] Schmied, T., et al. (2024). A large recurrent action model: xLSTM enables fast inference for robotics tasks. ArXiv
> >
> > [5] Yadav, S., et al. (2024). Audio xLSTMs: Learning self-supervised audio representations with xLSTMs. ArXiv

---

> > > ### Author Response · Authors · 2024-11-21
> > > **Any further questions?**
> > >
> > > Dear Reviewer,
> > >
> > > We hope to have resolved all your concerns. If there are any further comments from your side, we will be happy to address them before the rebuttal period ends. If there are none, then we would appreciate it if you could reconsider your rating.
> > >
> > > Regards,
> > >
> > > Authors

---

> > ### Comment · Reviewer_GoVE · 2024-11-22
> >
> > Thank you for the response. Although the author has elaborated on the contributions of xLSTM-Mixer and provided additional analyses to support their claims, the integrative innovation compared to the base xLSTM model is still insufficient, and the questions regarding motivation and innovation persist. The responses to the concerns raised about the experimental setup for varying input/output ratios, the model's behavior at different lookback lengths, and the omission of trend-seasonality decomposition, while present, do not fully address the concerns. Particularly, regarding the novelty and significant contributions of the model, the author needs to further clarify that xLSTM-Mixer is more than a trivial modification of xLSTM. Moreover, for the issue of the performance decline at certain lookback lengths, the author has updated experimental results but the explanation remains insufficient. Therefore, the rating remains unchanged.

---

> ### Author Response · Authors · 2024-11-22
>
> Dear Reviewer,
>
> Thank you for your response.
>
> We would like to clarify once more that there is no base model in xLSTMs.
>
> The xLSTMs [1] are a set of building blocks that can be used to construct various architectures, rather than a standalone model. To support this point, we refer to a statement from one of the follow-up works of the original xLSTM authors:
>
> > "The Extended Long Short-Term Memory (xLSTM) family [5] was recently introduced as a new architecture for language modeling." [2]
>
> The authors explicitly describe xLSTM as a family and a set of architectures for language modeling.
>
> Which is why finding viable combinations of building blocks (for example that the sLSTM block is a superior choice) is already a major contribution for the time modality. If we would decline such contributions we would today still use sigmoid or tanh activations in most of our deep networks.
>
> Additionally, in our manuscript, we conduct an ablation study on block selection (see Table 3) and further evaluate #10, where we focus solely on the pure xLSTM blocks. Moreover, xLSTMTime is the closest candidate that might be considered as an xLSTM (timeseries) base model, which we evaluate against in Tables 2, 5, and 6. However, as we stated in response Q4.1 to reviewer xNaw, we were unable to reproduce the originally claimed scores. Please refer to that response for more details.
>
> Furthermore, we contribute several additional essential components, such as time and variate mixing, key up- and down-projections, and the introduction of initial token embeddings, which, to the best of our knowledge, we are the first to explore. Additionally, we enhance the xLSTM and XAI community by demonstrating that attribution techniques can be transferred to forecasting and xLSTMs in general (again, to the best of our knowledge, we are the first to demonstrate this). Lastly, on average, we outperform all other methods by a large margin.
>
> We are confused if this is still not sufficient for an ICLR publication according to the reviewer.
>
>
>
> > "The responses to the concerns raised about the experimental setup for varying input/output ratios, the model's behavior at different lookback lengths, and the omission of trend-seasonality decomposition, while present, do not fully address the concerns."
>
> We also stated in our Global Response G1 that DLinear is a viable option and for nearly all tested scenarios it does not really matter which "backbone" to use. We are unsure what else we should test here. Furthermore, regarding "varying input/output ratios," could you please revisit our Q2 response and provide more concrete action points that we could address to satisfy your expectations? Your review was not entirely clear to us in this regard.
>
> [1] Maximilian Beck and Korbinian Pöppel and Markus Spanring and Andreas Auer and Oleksandra Prudnikova and Michael K Kopp and Günter Klambauer and Johannes Brandstetter and Sepp Hochreiter (2024). xLSTM: Extended Long Short-Term Memory. NeurIPS
>
> [2] Benedikt Alkin and Maximilian Beck and Korbinian Pöppel and Sepp Hochreiter and Johannes Brandstetter (2024). Vision-LSTM: xLSTM as Generic Vision Backbone. ArXiv

---

> > ### Author Response · Authors · 2024-11-27
> > **Did we resolve your concerns?**
> >
> > Dear Reviewer,
> >
> > We hope to have clarified your outstanding concerns. Let us know if there are any further questions. If no, it will be great if you could reconsider your score.
> >
> > Regards,
> >
> > The Authors

---

> > > ### Author Response · Authors · 2024-12-02
> > >
> > > Dear Reviewer GoVE,
> > >
> > > Thank you for the constructive discussion. We hope to have clarified all remaining questions.
> > >
> > > As the reviewing period will soon close, we would kindly ask you to reconsider your currently unchanged score after our extensive clarifications and significant additions. We specifically addressed your constructive comments in the revised manuscript and comments.
> > >
> > > Regards,
> > >
> > > The Authors

---

> > ### Comment · Reviewer_GoVE · 2024-12-03
> >
> > Thanks for your response and efforts you have made to address these concerns. We appreciate the additional analyses to support your arguments. In light of this, we have increased the presentation and contribution scores.
> >
> > - However, we would like to bring to your attention that the concern regarding the exploration of different input/output ratios remains unresolved. "Explore different input/output ratios" refers to the proportion between the length of lookback window and the length of forecasting horizon. This issue has not been adequately addressed, and the title of Figure 5, "xLSTM-Mixer effectively learns cross-variate patterns," does not constitute a response to this concern.
> >
> > - The contribution of this paper does not lie in the xLSTM itself, and finding viable combinations of building blocks cannot be considered a 'major' contribution. At the meanwhile, the authors have not tested other potential blocks to validate the reliability of the framework.
> >
> > - Additionally, we have noticed that the order of the figures in the manuscript is still confusing and requires improvement. Clarity in the presentation of your work is crucial for the reader's understanding and the overall impact of your research.

---

> > > ### Author Response · Authors · 2024-12-03
> > > **Addressed Final Concerns**
> > >
> > > Dear Reviewer GoVE,
> > >
> > >
> > > Thank you for your response and for raising your meta scores. We are confident that we have now addressed all remaining concerns:
> > >
> > > 1. In our initial response, the reference to Figure 5 referred to the original version of the paper. **In the revised PDF, the experiments you requested are now shown in Figure 6**. We apologize for this misunderstanding. As you suggested, we have expanded the lookback length $T$ to include values up to 2048. In total, we evaluated **$7 \times 4 = 28$ combinations** of input and output ratios, for **six models**, each with **three different seeds**. We believe this is sufficient.
> > > **Other reviewers agree and think our evaluation in that regard is sufficient.**
> > >
> > > 2. As per your suggestion, **we also investigated the other central block (namely mLSTM)** instead of the sLSTM in our ablation study (see Table 3; column \#2). Regarding other blocks, just like Chen et al. [1] is based on MLPs (and *only* evaluated on those), our method is based on the powerful xLSTM framework, where we utilize the built-in cell mixing capabilities of the sLSTM block. While evaluating completely different architectures remains an exciting opportunity for future work, we did evaluate, in our extensive ablation study, scenarios where parts of the mixing were omitted or swapped out. We demonstrate that retaining these components is indeed beneficial. **Other reviewers agree and think our evaluation in that regard is sufficient.**
> > >
> > > 3. We will revisit the figure ordering specifically in Section 4.2 for the camera-ready version. This will ensure that **the many extensions added during the rebuttal are integrated into a coherent structure**.
> > >
> > > Thank you for the helpful exchange and for now reconsidering your rating, now that all your concerns have been fully addressed.
> > >
> > > Regards,
> > > The Authors
> > >
> > >
> > >
> > > [1] Chen et al., TSMixer: An All-MLP Architecture for Time Series Forecasting. TMLR (2023)

---

### Official Review · Reviewer_xNaw · 2024-11-08

**Soundness:** 2
**Presentation:** 2
**Contribution:** 2
**Rating:** 6
**Confidence:** 3

**Summary:**

The paper introduces xLSTM-Mixer, a new model for forecasting complex time series data with multiple variables. It combines the xLSTM architecture with multi-view processing and variate mixing. The model works in three stages: first, it creates an initial forecast, then it uses xLSTM blocks to combine time and variable patterns, and finally, it merges two different forecast views for results in most cases beating baseline models. xLSTM-Mixer addresses challenges associated with Transformer models, such as high computational costs and data requirements for long sequences.

**Strengths:**

**Originality**
xLSTM-Mixer combines temporal sequences, joint time-variate information, and multi-view perspectives to handle multivariate time series complexities, capturing temporal and cross-variate patterns for accurate predictions, it builds on prior work, but combines it in an intricate way. The work enhances the xLSTM architecture, however limiting itself to the sLSTM.

**Quality**
The work is written with enough clarity, related work is addressed and the paper is fairly easy to understand. The graphical presentation of the model allows for easy following.

**Significance**
Authors do not claim state-of-the-art performance, however, they do beat the current baselines to which they compare themselves. It is the second work relating to the use of xLSTM to time series, and authors raise concerns about the first one, those concerns must be backed in the future.

**Weaknesses:**

**Weaknesses**

1. The authors conduct experiments on only a subset of datasets, specifically focusing on a subset of the Informer long-horizon forecasting datasets and omitting the Exchange dataset, which would add considerable value if included. The reviewers would appreciate a deeper analysis of why the XLSTM-Mixer performs better on some datasets and worse on others, particularly through an exploration of cross-correlation between variables and model performance.

2. The ablation study, conducted on only two datasets, lacks breadth. Expanding this experiment across additional datasets would significantly strengthen the findings.

3. The experiments do not address a critical aspect: the model's sensitivity to the order of variates, as it is not permutation invariant in this regard.

**Questions:**

1. The term "Mix View" in Table 3, line 332, is unclear and challenging to interpret. Could you clarify if this refers to the performance of channel mixing?

2. The dataset selection could be expanded to include the Exchange from the  (Informer long-horizon forecasting datasets) datasets and more extensive ablation studies across additional datasets to provide a thorough evaluation of the xLSTM-Mixer’s performance, including an exploration of cross-correlation between variables and model effectiveness on different datasets or synthetic ones.

3. Consider incorporating synthetic data to facilitate verification of learnable concepts and addressing the model’s sensitivity to the order of variates, as it lacks permutation invariance.

4. I would highly encourage you to back the claims of challenges in the reproducibility of prior works and why trend-seasonality decomposition is redundant.

---

> ### Author Response · Authors · 2024-11-20
>
> Thank you for your constructive feedback on our paper. We appreciate the time and effort you have invested in reviewing our work. Your comments provide valuable insights that will help us enhance the quality and clarity of our paper.
>
> We're also glad that the clarity of our presentation and visual model representations contributed to your understanding of our approach. Your acknowledgment of our contribution to advancing the xLSTM architecture in time series is motivating, and we aim to continue building on this foundation.
>
> Regarding your suggestions for improvement and specific questions, we address them below in detail:
>
> - **W1 (deeper analysis of forecasting results)**: We happily expand on our findings from Sec. 4.1. Firstly, xLSTM-Mixer provides SOTA results for most settings. In some cases, particularly on Traffic, it greatly improves the MAE while not surpassing TiDE and TimeMixer on the MSE. We conjecture that this follows from the strong regularization of our mixing operations, which does constrain what the model can learn and might be slightly adverse when capturing high-frequency values and outliers (note that the minimal Hurst exponent of Traffic is very low, as shown in the update Tab. 1). Similarly, as sLSTM-Mixer strides over the variates, some particularly long-running connections have to be learned, which is rather challenging. Our response to W3 provides additional insights.
>
> - **W2 (extending the ablation study)**: We have already performed the ablation study on two datasets, over four horizons each, and by initializing with three seeds (i.e., by training 24 models per variation, resulting in a total of 288 models). We do agree that the ablation could be meaningfully extended by adding more model variations; however, we do not believe that adding more datasets will provide any significant new insights. We, therefore, provide additional insights into normalization based on trend-seasonality decomposition and replacing sLSTM with mLSTM blocks. These additions have been included in our ablation in Table 3 in the revised manuscript and are also addressed in G1.
>
> - **W3/Q2/Q3 (variations of variate orders)**: The model is indeed dependent on the ordering of variates. We empirically found this to not be a significant limitation, as the standard ordering in the dataset source was already highly effective for forecasting. We, therefore, leave models like bi-directional xLSTMs to future work (cf. ll. 543ff). However, we conducted an attribution study to verify that the model indeed effectively captures cross-variate information. We achieved this without synthetic data, ensuring the findings are truthful to the models used in the wider evaluation. We also added a new experiment that investigates the impact of variate ordering. Please also see the revised manuscript in Appendix 5.
>
> - **Q1 (clarification on "Mix View")**: Thank you for pointing this out. The term refers to multi-view mixing described in Section 3.3. We updated Table 3 to clarify this.
>
> - **Q2 (evaluation on yet more datasets)**: We followed recent related work [1,2,3,4], which do not evaluate on Exchange (and except of [2] neither ILI), as the remaining datasets already showcase the model capabilities. However, we agree that the long/trend forecasting of ILI is an interesting evaluation opportunity due to its limited data points and challenging trend forecasting. This is further discussed in our response to FwwC, Q4. We have added the evaluation of xLSTM-Mixer and all other methods on ILI. The results can be found in the revised manuscript in Tables 2 and 6.
>
> - **Q4.1 (reproducibility challenges)**: We tried to address the reproducibility issues as politely as possible in lines 515-520 of our submission. Unfortunately, the work by Alharthi & Mahmood [5] does not provide any details about the chosen hyperparameters or the range used in their hyperparameter search. Their code also lacks proper configuration settings, making it difficult to replicate their experiments without modifying the code directly. We contacted the authors, but they only shared a very minimal learning rate search strategy without providing detailed results. Critical factors, such as whether an mLSTM or sLSTM block should be used, were not specified, despite their claim that the block selection is crucial for each dataset. Consequently, we attempted to find suitable configurations ourselves but were unable to achieve the reported performance. Therefore, for fairness, the reported statistics in Tables 2 and 6 are sourced directly from the original paper.
>
> - **Q4.2 (trend-seasonality decomposition)**: Thank you for this suggestion to extend the rigor beyond describing findings from initial model analysis. As laid out in W2 and G1, we are happy to provide this as an additional thorough ablation.
>
> Thank you again for your valuable comments, and we are happy to answer any remaining questions you may have.

---

> > ### Author Response · Authors · 2024-11-20
> > **References**
> >
> > [1] Wang, S., et al. (2024). TimeMixer: Decomposable multiscale mixing for time series forecasting. ICLR
> >
> > [2] Nie, Y., et al. (2023). A time series is worth 64 words: Long-term forecasting with transformers. ICLR
> >
> > [3] Das, A., et al. (2023). Long-term forecasting with TiDE: Time-series dense encoder. TMLR
> >
> > [4] Chen, S.-A., et al. (2023). TSMixer: An all-MLP architecture for time series forecasting. TMLR
> >
> > [5] Alharthi, M. and Ausif M. (2024): xLSTMTime: Long-Term Time Series Forecasting with xLSTM. MDPI AI

---

> > > ### Author Response · Authors · 2024-11-21
> > > **Any further questions?**
> > >
> > > Dear Reviewer,
> > >
> > > We hope to have resolved all your concerns. If there are any further comments from your side, we will be happy to address them before the rebuttal period ends. If there are none, then we would appreciate it if you could reconsider your rating.
> > >
> > > Regards,
> > >
> > > Authors

---

> > > > ### Comment · Reviewer_xNaw · 2024-11-26
> > > >
> > > > Dear authors,
> > > > Thank you very much for the reply. The changes you have introduced in my opinion improve readability and positively add to the paper. I maintain the positive score of 6, since I think the lack of permutation invariance is still a hurdle and is not fully addressed by the experiment added in A.5, by the small subset of permutations.
> > > > I wish you all the best.
> > > > Reviewer XNaw:)

---

> > > > > ### Author Response · Authors · 2024-11-26
> > > > > **Scope of the Evaluation**
> > > > >
> > > > > We are grateful for your help in improving the readability of the paper, and we believe that the newly added experiments have further enhanced its quality. We are also pleased that we have addressed all other points to your satisfaction and appreciate your positive score. In light of the borderline decision, we would kindly ask what would be required to fully convince you to increase your score to "accept."
> > > > >
> > > > > We believe the current set of variate ordering experiments is sufficient, as the total number of experiments was already significant. Specifically, we ran experiments on three datasets (including the addition of the "Electricity" dataset since our last response), four horizons, as is common practice, four different permutations, and each on three seeds to account for any potential variability. This results in a total of 144 new experiments for this table alone.
> > > > >
> > > > > Although not reported here, the standard deviations observed during our experiments were very small, which aligns with the results now presented in Appendix A.6. This further underscores the reliability of our findings. However, if the reviewer holds a different opinion, we are open to extend our evaluation even further.
> > > > >
> > > > > Thank you once again for your support and the fruitful discussion.
> > > > >
> > > > > Regards,
> > > > >
> > > > > Authors

---

### Author Response · Authors · 2024-11-20
**Global Response**

We thank all the reviewers for their time in reading and reviewing our manuscript and for their valuable comments, which are helpful for us to further improve our paper.

We have made every effort to address all opportunities for improvement and questions by providing detailed clarifications and requested results.

All revisions made to the manuscript are color-coded for easier detection:
- XNaw: violet
- GoVE: orange
- 9duf: teal
- FwwC: red
- cZwn: brown
- Revisions spanning multiple Reviewers: cyan

---

> ### Author Response · Authors · 2024-11-20
> **G1. Incorporating trend-seasonality decomposition**
>
> We employed normalization akin to NLinear (Zeng et al., 2023) for the initial forecast (cf. Sec. 3.1). As pointed out by several reviewers (XNaw and GoVE), we investigated whether trend-seasonality decomposition is viable as well. We extended the ablation study and updated the PDF with an extended Table 3. Comparing the baseline with NLinear in column #1 to the DLinear variant in column #4, we can see that NLinear is indeed mostly superior. However, DLinear is a viable option to improve the MSE (preventing high individual errors) in ETTm1. The slight improvement by employing normalization based on a moving average is plausible, given that ETTm1 exhibits significantly more high-frequency dynamics (see Figure 2).

---

> ### Author Response · Authors · 2024-11-20
> **G2. Learning cross-variate relationships and the role of variate orderings**
>
> Multiple reviews suggested further investigating the role of the specific order of the variates. We analyze this from two distinct perspectives:
>
> (1) To show which cross-variate relationships are learned by xLSTM-Mixer, we added experiments in ll 408-419, "Learning Cross-Variate Patterns". Results on more datasets can be found in Appendix A.3. In summary, xLSTM-Mixer effectively learns relationships between variates in the provided order.
>
> (2) To assess to which degree variate ordering influences forecasting effectiveness, we conducted experiments on four different randomized orders and compared them with the standard ordering provided by the dataset source. As the results in Appendix A.5 show, the specific ordering does play some role in forecasting performance. Yet, the standard ordering already permits highly effective forecasting.
>
> Extending xLSTM-Mixer with bi-directionality might further unlock hidden potential. Yet, we leave this open for future work to explore, as xLSTM-Mixer is already highly effective at forecasting and provides a valuable new state-of-the-art for the forecasting community.

---

> ### Author Response · Authors · 2024-11-20
> **G3. Computational efficiency**
>
> As noted by 9duf (W2), GoVE (W4), and FwwC (Q3), investigating the efficiency of xLSTM-Mixer in terms of computation time and memory requirements would help evaluate its suitability for applications. We updated the PDF to contain now a "Model Efficiency" section in ll. 421-477. In summary, xLSTM-Mixer scales extremely favourable in the lookback length compared to the other models while consuming significantly less memory.

---

### Meta-Review · Area_Chair_KKv7 · 2024-12-21

**Metareview:**

This paper has been evaluated by 5 knowledgeable reviewers. Their opinions varied: 1 straight acceptance, 1 marginal acceptance, 1 marginal rejection and 2 straight rejections. The authors provided extensive rebuttals which addressed some of the reviewers’ complaints. The paper considers long-term time series forecasting using xLTM as a mixer adapted to capture both the temporal and cross-channel information. It shows good empirical performance but leaves some issues unaddressed. Reviewers brought up the lack of analysis of the impact of relative lengths of the observed input time series and the forecast horizon, they also wished for more comparisons against more data and a broader range of relevant algorithmic baselines, to comprehensively assess the relative utility of the proposed method. Reviewers also pointed out incremental novelty of the presented methods. This promising work will need a few substantial upgrades to be accepted for publication at ICLR.

**Additional Comments On Reviewer Discussion:**

Two reviewers engaged in a conversation with the authors, discussing details of the rebuttal. There was no formal discussion of the paper among the reviewers, but it seems that they have considered the information provided by their colleagues while arriving at their final assessments.

---

### Decision · Program_Chairs · 2025-01-22

Reject